# Impact of time-restricted feeding on metabolic health and adipose tissue metabolism in aged female mice with high-fat diet-induced obesity

Theresa Bushman, Hongming Su 🆔 and Xiaoli Chen 🆔

*Department of Food Science and Nutrition, University of Minnesota-Twin Cities Minnesota, St. Paul, USA*

Handling Editors: Karyn Hamilton & Josiane Broussard

The peer review history is available in the Supporting Information section of this article (https://doi.org/10.1113/JP289464#support-information-section).

**Abstract figure legend** A 10 week time-restricted feeding (TRF) regimen was implemented in aged female mice following 12 weeks of high-fat diet (HFD) feeding. TRF partially reversed HFD-induced weight and fat mass gain, reduced adipocyte size and increased size heterogeneity in white adipose tissue. It also enhanced energy expenditure, lowered RER (particularly during the light phase), decreased fasting blood glucose and reduced hepatic lipid accumulation. At the molecular level TRF promoted metabolic remodelling in adipose tissue, including upregulation of genes related to adipogenesis and lipid cycling, with depot-specific changes in mitochondrial oxidation and circadian rhythm gene expression.

**Abstract**  Ageing increases the risk of obesity and related metabolic diseases, emphasizing the need to understand how dietary interventions influence metabolism and metabolic health in older populations. This study aimed to investigate the impact of time-restricted feeding (TRF) on energy balance, adipose tissue metabolism and overall metabolic health in aged female mice with high-fat diet (HFD)-induced obesity. A 10-week TRF regimen was implemented in aged female mice following 12 weeks of HFD exposure. Mice were either maintained on HFD *ad libitum* (HFD-AL)

**Theresa Bushman**, PhD, RD, earned her doctorate in nutrition from the University of Minnesota and is a paediatric clinical dietitian specializing in weight management. Her research has focused on time-restricted feeding (TRF) and its role in metabolism, ageing, and health, including how dietary timing influences weight, fat distribution and metabolic function in pre-clinical models. Dr Bushman is committed to advancing nutrition research that bridges science and clinical practice to support healthier ageing.

The Journal of Physiology

or subjected to TRF with HFD access restricted to a 10 h daily feeding window (HFD-TRF). Glucose and insulin tolerance tests, meal pattern and indirect calorimetry were measured during the regimen. We showed that TRF partially reversed HFD-induced weight gain and fat mass accumulation. In white adipose tissue TRF reduced average adipocyte size and increased the heterogeneity in adipocyte size distribution. TRF also led to increased $VO_2$ and $VCO_2$, along with a decreased respiratory efficiency ratio (RER) compared to the HFD-AL group, particularly during the light phase. Meal pattern analysis showed increased meal frequency during the feeding window in HFD-TRF mice relative to HFD-AL. Additionally TRF lowered fasting blood glucose and reduced liver lipid accumulation. At the molecular level TRF induced significant metabolic adaptations in adipose tissue, including upregulation of genes involved in adipogenesis and lipid cycling, as well as depot-specific alterations in mitochondrial oxidation and circadian rhythm gene expression. In conclusion TRF promotes beneficial metabolic adaptations and may serve as an effective dietary strategy to improve metabolic health in aged females.

(Received 11 June 2025; accepted after revision 17 December 2025; first published online 5 January 2026)

**Corresponding author** X. Chen: Department of Food Science and Nutrition, University of Minnesota-Twin Cities, St. Paul, MN, USA.    Email: xlchen@umn.edu

## Key points

- Time-restricted feeding (TRF) reduced body weight and fat mass, lowered blood glucose and decreased lipid accumulation in the liver.
- TRF also changed energy fuel utilization, increased metabolic activity of adipose tissue and altered the size and function of fat cells.
- Altered meal timing can trigger beneficial metabolic changes and suggests that TRF may help protect against obesity-related diseases during ageing.

## Introduction

The rising prevalence of obesity and diabetes has intensified research into how dietary patterns affect energy balance, adipose tissue metabolism, insulin resistance and overall metabolic health. With age the risk of developing obesity and related metabolic diseases, including type 2 diabetes and cardiovascular disease, increases. Therefore identifying effective dietary interventions to control obesity during ageing is critical for reducing the burden of chronic metabolic diseases. Among these strategies intermittent fasting (IF), particularly time-restricted feeding (TRF), which limits the daily eating window and helps restore circadian rhythms disrupted by modern dietary patterns, has emerged as a promising approach for improving metabolic health without necessarily reducing caloric intake (de Cabo & Mattson, 2019; Varady et al., 2022; Zeb et al., 2021). TRF has shown to benefit weight loss, blood pressure regulation, insulin sensitivity and oxidative stress in humans (Cienfuegos et al., 2020; Sutton et al., 2018). These findings highlight TRF's potential to reduce the prevalence of obesity and type 2 diabetes, warranting further investigation.

Adipose tissue plays a central role in regulating energy metabolism. Recent studies have demonstrated that adipose tissue is highly responsive to the ageing environment (Nguyen & Corvera, 2024; Pálovics et al., 2022; Schaum et al., 2020), suggesting that its dysfunction contributes to organismal ageing and the development of age-related diseases. Chronic low-grade inflammation stemming from adipose tissue is a key factor in this process, occurring both in obesity and with age. When age-related, this phenomenon, known as 'inflammaging', is characterized by elevated pro-inflammatory cytokines and activation of inflammatory pathways (Mau & Yung, 2018). Mitochondrial dysfunction, common in both obesity and ageing, along with inflammaging, increases susceptibility to metabolic disorders such as insulin resistance and atherosclerosis (Amorim et al., 2022). Therefore understanding how TRF influences adipose tissue metabolism and function in an aged, diet-induced obesity model is essential for developing strategies to combat age-associated metabolic dysfunction.

Adipose tissue is generally classified into brown adipose tissue (BAT) and white adipose tissue (WAT), which differ in morphology, gene expression and metabolic functions. Brown adipocytes are characterized by multiple small lipid droplets (15–60 μm) and contribute to energy expenditure through thermogenesis (Cedikova et al.,

2016; Giralt & Villarroya, 2013; Peirce et al., 2014). In contrast white adipocytes contain a single large lipid droplet (25–200 μm) (Cedikova et al., 2016). In WAT insulin stimulates the storage of energy by promoting fatty acid and glucose uptake leading to triglyceride synthesis (Kahn & Flier, 2000). Under energy-deficient conditions stored triglycerides undergo lipolysis, releasing fatty acids and glycerol for use by tissues. Excess caloric consumption leads to adipose tissue expansion via both hyperplasia and hypertrophy mechanisms (White, 2023). Studies of high-fat diet (HFD)-induced adipose tissue expansion have demonstrated that both adipocyte hyperplasia and hypertrophy occur during early HFD exposure (Jo et al., 2009, 2010). However with prolonged HFD feeding adipocyte hypertrophy continues, whereas adipocyte number declines (Jo et al., 2010), indicating impaired adipogenesis over time. Weight loss has been shown to reverse adipocyte hypertrophy, improving adipocyte physiology in obese individuals (Varady et al., 2009).

Ageing is also associated with a natural decline in basal metabolic rate (BMR), which is the energy expenditure needed to maintain basic physiological functions at rest (Henry, 2000; Pataky et al., 2021), due to reductions in lean body mass and changes in hormonal regulation (Henry, 2000; Pataky et al., 2021). This leads to gradual increases in adiposity and a redistribution of fat from subcutaneous to visceral depots. Additionally research indicates that basal fat oxidation and maximal oxidative capacity are decreased in older obese individuals (Solomon et al., 2008), and older mice exhibit lower energy expenditure and $VO_2$ max during physical activity compared to younger mice (Houtkooper et al., 2011). Ageing also impairs metabolic adaptability in response to nutritional challenges, further increasing the risk of metabolic disorders (Manolopoulos et al., 2010).

TRF has been shown to influence adipose tissue metabolism by modulating adipocyte size, distribution and function (Antoni et al., 2017; Bushman et al., 2023; Chaix & Zarrinpar, 2015). The mechanisms of TRF are multifaceted, affecting various metabolic pathways within adipose tissue. Many TRF studies have used mice fed long-term high-fat diet to model human dietary patterns, especially the Western diet. Young male mice fed a regular chow diet consume approximately 20% of their daily food intake during the light cycle, whereas HFD-FED mice consume about 30% during this period, likely due to of circadian rhythm disruption caused by the HFD (Kohsaka et al., 2007). Although multiple studies have reported the metabolic benefits of TRF in young male mice, with evidence suggesting that each adipose depot responds differently to a TRF regimen (Bushman et al., 2023), there is limited understanding of its depot-specific effects on adipose tissue metabolism in aged female mice with diet-induced obesity. Importantly one study using a postmenopausal female mouse model

fed a HFD demonstrated that TRF led to rapid weight loss, improved glucose tolerance and reduced hepatic lipid accumulation, accompanied by a shift in hepatic metabolism from gluconeogenesis towards ketogenesis (Chung et al., 2016). This is particularly relevant because postmenopausal women experience unique metabolic risks, including increased visceral adiposity, insulin resistance and susceptibility to non-alcoholic fatty liver disease (Chung et al., 2015). Yet most preclinical metabolic studies have relied on young male rodents, overlooking sex- and age-related differences in disease pathophysiology and therapeutic response. This gap in research limits the translational relevance of finding for older women, who represent a high-risk and growing patient population. Therefore the aim of this study was to investigate the effects of TRF on energy expenditure, metabolic health outcomes and adipose tissue metabolism across three distinct fat depots in aged female mice with HFD-induced obesity.

## Methods

### Ethical approval

All procedures were approved by the University of Minnesota Institutional Animal Care and Use Committee (IACUC 2102A38852) and followed NIH guidelines for laboratory animal care. Animal killing procedure: anaesthetics were not used for animal killing in this study. Instead carbon dioxide ($CO_2$) inhalation was employed in accordance with protocols approved by the IACUC at the University of Minnesota. Compressed $CO_2$ gas was supplied from cylinders and regulated using a pressure-reducing regulator and flow meter. Animals were placed in a euthanasia chamber, and 100% $CO_2$ was introduced at a fill rate of 30%–70% of the chamber volume per minute to displace ambient air. Unconsciousness was confirmed by the absence of respiration and the presence of corneal opacification. $CO_2$ flow was maintained for at least 1 min beyond the cessation of respiration to ensure complete euthanasia. After this cardiac puncture was performed for blood collection. When necessary cervical dislocation was used as a secondary method to confirm death and facilitate tissue collection.

### Animal study

Animals were housed at 22°C in a specific pathogen-free facility at the University of Minnesota. The study included 27 aged (14–18-month-old) female C57BL/6 mice (The Jackson Laboratory, Bar Harbor, ME, USA). Twenty mice were fed a HFD for 12 weeks to induce obesity, whereas seven mice remained on an *ad libitum* normal chow diet

(control). After 12 weeks HFD-fed mice were divided into two groups: 10 continued on *ad libitum* HFD (HFD-AL) group and 10 were switched to time-restricted feeding (HFD-TRF) group. In addition 11 female mice at 3 months of age were enrolled in a middle-aged study. These consisted of four chow-fed controls, four HFD-AL and three HFD-TRF. After 12 weeks of HFD feeding these mice reached 6 months of age and were either maintained on *ad libitum* HFD (HFD-AL) or switched to time-restricted feeding (HFD-TRF) for additional 10 weeks. An experimental timeline for both studies is shown in Fig. 1. The AL groups had constant access to food, and the TRF group had access to food 10 h/day during the active (dark) period. The mice were housed in groups of 3–4 per cage, with water *ad libitum* and in 12 h light/dark cycles with lights off from 12–24 ZT (8 PM–8 AM). The HFD-TRF group had food available from 8:30 PM–6:30 AM. The high-fat diet provided to the mice was a 60% HFD (Bio-Serv: F3282), and the normal chow was provided by the animal facility (Envigo: 2918). According to the manufacture (https://www.bio-serv.com/product/HFPellets.html) the HFD contains 20.5% protein, 36% fat and 35.7% carbohydrate compared to the normal diet, which contains 20.5% protein, 7.2% fat and 61.6% carbohydrate. Regarding the type of fat the diet contains lard as its primary fat source. Glucose tolerance test (GTT) was conducted on week 6, and insulin tolerance test (ITT) was conducted on week 7. The mice were placed in metabolic cages at week 8. Food intake was measured over three consecutive days. After 10 weeks of the dietary intervention all three groups were killed following a 16 h fast. The mice were killed, and blood was collected through cardiac puncture. Brown adipose tissue, inguinal adipose tissue, epididymal adipose tissue, retroperitoneal fat, kidneys, liver and muscle were collected and weighed.

Tissue section from brown adipose tissue and liver was used for histological analysis, whereas small portions of inguinal and epididymal adipose tissue were used for both histological analysis and fat cell sizing, and the remaining tissue was snap-frozen in liquid nitrogen and stored for later analysis.

### Indirect calorimetry and body weight

Body composition (fat mass and fat-free mass, both in grams, Echo MRI 3-in-1, Echo Medical System) was measured. Mice were individually caged at week 8 of the TRF regimen, and food intake was measured over a 5-day period using a Biodaq food intake monitoring system (Research Diets Inc., New Brunswick, NJ). The final 2 days of food intake monitoring were averaged to determine total food intake in grams/day. Indirect calorimetry (oxygen consumption ($VO_2$), carbon dioxide production ($VCO_2$), heat, respiratory efficiency ratio (RER) and activity) was measured over a 3-day period using the Oxymax Comprehensive Lab Monitoring System (Columbus Instruments, Columbus, OH, USA). The final light and dark cycles were used to statistically determine differences in energy expenditure between experimental groups.

### Fat cell sizing

Adipose tissue was obtained from inguinal and epididymal WAT of mice fed control or HFD with or without TRF. Tissue samples (25–30 mg) were immediately fixed in 12 ml of the 2% osmium tetroxide solution in collidine buffer in a Wheaton vial (SPI-Chem no. 986704) and incubated in a water-bath at 37°C

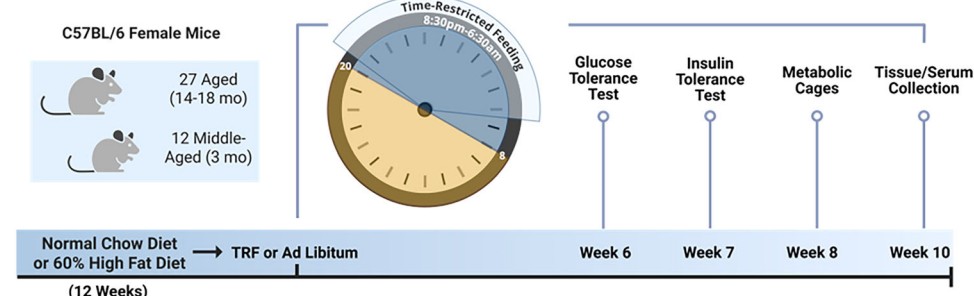

**Figure 1. Experimental timeline. In the aged cohort 27 aged female mice (14–18-month old) were used**
Seven mice remained on an *ad libitum* normal chow diet (control), whereas 20 were fed a high-fat diet (HFD) for 12 weeks to induce obesity. The HFD-fed mice were then divided into two groups: *ad libitum* HFD (HFD-AL, *n* = 10) or time-restricted feeding (HFD-TRF, *n* = 10). In a parallel study 11 middle-aged female mice (3-month old) were assigned to chow (*n* = 6), HFD-AL (*n* = 3) or HFD-TRF (*n* = 4). After 12 weeks of HFD they reached 6 months of age and continued on HFD-AL or HFD-TRF for an additional 10 weeks. Mice were housed 3–4 per cage under 12 h light/dark cycles (lights off 8 PM–8 AM), with water *ad libitum*. TRF groups had food access for 10 h during the dark phase (8:30 PM–6:30 AM). At weeks 6–7 of TRF GTT and ITT were performed (6 mice per group); indirect calorimetry and meal pattern analysis were performed at week 8 (6 mice per group). At week 10 mice were killed for blood and tissue collection.

for 48 h, as described previously (Hirsch & Knittle, 1970; McLaughlin et al., 2007). Collidine buffer was prepared from a 4°C stock collidine buffer of 0.2 M 2,4,6-trimethyl-pyridine (C-0505; Sigma Chemical Co) dissolved in distilled water. Subsequently the contents of the vial were washed out using a 25 μm filter to catch the fixed cells, and the container was rinsed three more times. After a final rinse using 0.9% saline, the material was transferred into a 250 μm filter using a squirt bottle filled with 0.9% saline. A gentle rub on the 250 μm mesh was applied to crush any large chunks and then rinsed again with 0.9% saline. The procedure was repeated to collect the cells. The end volume did not exceed more than one conical tube (50 mL). Samples were analysed with an optical particle analyse (model: Microtrac SIA), which is a particle characterization tool. The Bluewave can measure particles in the size range 50 nm to 2800 μm. Each sample was measured in duplicate. Figures were generated using GraphPad Prism version 9.5.1 for Windows (GraphPad Software, San Diego, CA, USA).

### Glucose and insulin tolerance test

GTT was conducted at week 6 of the TRF regimen. The mice were fasted for 16 h. Body weight was recorded. Bolus of glucose (0.75 g/kg body weight) was prepared from a filter-sterilized D-glucose solution (300 mg/ml in saline) and administered intraperitoneally at 2.5 μl per gram body weight. Fasting blood glucose was measured from the tail vein prior to glucose intraperitoneal injection. Blood glucose was measured at 15, 30, 45, 60, 90, 120 min intervals. ITT was conducted at week 7 of the TRF regimen. The mice were fasted for 4 h prior to injection. Body weight was recorded. A bolus of insulin (0.75 U/kg body weight) was prepared from a diluted stock solution of regular human insulin (Humulin R; Lilly NDC 0002-8215-01). Fasting blood glucose was measured prior to insulin intraperitoneal injection. Blood glucose was measured at 15, 30, 45, 60, 90, 120 min intervals.

### Haematoxylin and eosin staining of tissues

Tissues were fixed in 10% neutral buffered formalin (VWR International, LLC, Radnor, PA, USA) for 2–3 days, dehydrated in 70% ethanol solutions for 3–5 days and processed for embedding in paraffin. Tissue samples were haematoxylin and eosin (H&E) stained using a standard protocol at the University of Minnesota Histology Core. Briefly after deparaffinization and rehydration tissues were sectioned with 5–6 μm thickness and stained in haematoxylin for 1 min and rinsed with distilled water. After haematoxylin staining tissues were counterstained with eosin solution for 1 min followed by dehydration through 95% and 100% EtOH and xylene clearance. At last

the tissue sections were mounted with resinous mounting medium. Images were then captured using a Leica DM IL microscope.

### Quantitative real-time PCR

Total RNA was extracted from frozen tissue using TRIZOL reagent (Invitro, Carlsbad, CA, USA) and treated with DNAase to remove genomic DNA prior to cDNA synthesis using Superscript II reverse transcription kit (Invitrogen). Real-time quantitative PCR was conducted using FastStart Universal SYBR Green Master (Rox) (Roche) with a QuantStudioTM 3 Real-Time PCR System (Applied Biosystem, Foster City, CA, USA). The $\Delta\Delta$Ct method was used to calculate mRNA expression. For quantification TATA-box binding protein (*Tbp*) mRNA served as an endogenous control within inguinal and brown adipose tissue. The primer sequences for amplifying the target genes are shown in Table 1.

Forward (F) and reverse (R) primer sequences for each gene are listed along with gene name and gene symbol. Primers were validated for specificity using melt curve analysis.

### Serum analyses

Serum triglyceride level was determined using Triglycerides Enzymatic Assay Kit (Stanbio Laboratory, Boerne, TX, USA; #2100430). Serum-free fatty acids and $\beta$-hydroxybutyrate levels were determined using a Free Fatty Acid Quantitation Kit (Sigma-Aldrich, # MAK044) and a $\beta$-Hydroxybutyrate Assay Kit (Sigma-Aldrich, # MAK041) according to the manufacturers' instructions.

### Statistical analysis

Results are expressed as mean ± SD. Data from aged and middle-aged female mice were analysed using Student's *t* test when comparing two groups, and one-way ANOVA when comparing more than two groups, using GraphPad Prism (version Prism 9.4.1). Statistical significance was defined as a *P*-value $< 0.05$.

## Results

### Effect of TRF on body weight and adiposity in aged female mice

To investigate the beneficial effects of TRF on metabolic health in aged mice we placed mice at the age of 14–18 months on a HFD for 12 weeks followed by either HFD-AL or HFD-TRF for an additional 10 weeks. During the TRF period the HFD-AL group showed no significant differences in weight gain compared to the

**Table 1. Primer sequences used for quantitative real-time PCR**

| Gene | Gene symbol | Forward primer | Reverse primer |
|---|---|---|---|
| Uncoupling protein 1 | *Ucp1* | TAATGACTGGAGGTGTGGCAGTGT | TGTTGACAAGCTTTCTGTGGTGGC |
| Transcription factor A, mitochondrial | *Tfam* | CACTGGGAAACCACAGCATACAG | GGACATCTGAGGAAAAGCCTTGC |
| Oestrogen-related receptor alpha | *Errα* | CCAGACAGCAGCCTCAAAAAC | GATAGGGACCGAACACAGATCCT |
| Peroxisome proliferator-activated receptor gamma coactivator 1-alpha | *Pgc-1α* | ACCGTAAATCTGCGGGATGATGGA | AGTCAGTTTCGTTCGACCTGCGTA |
| Stearoyl-coenzyme A desaturase 1 | *Scd1* | CACTGAATGCGAGGGTTGGTTGTT | TCCTTTCAGCAGCACTGTACCACT |
| Elongation of very-long-chain fatty acids protein 5 | *Elovl5* | GGTGGCTGTTCTTCCAGATT | CCCTTCAGGTGGTCTTTCC |
| Carnitine palmitoyl transferase 1 | *Cpt1* | CCTCCCTGGGCATGATTG | ACGCCACTCACGATGTTCTTC |
| ATP synthase F1 subunit beta | *Atp5b* | GCAAGGCAGGGACAGCAGA | CCCAAGGTCTCAGGACCAACA |
| Cell death-inducing DNA fragmentation factor alpha-like effector A | *Cidea* | TGCTCTTCTGTATCGCCCAGT | GCCGTGTTAAGGAATCTGCTG |
| Cytochrome c oxidase subunit IV | *COXIV* | ATGTCACGATGCTGTCTGCC | GTGCCCCTGTTCATCTCGGC |
| Adipose triglyceride lipase | *Atgl* | TGTGGCCTCATTCCTCCTAC | TCGTGGATGTTGGTGGAGCT |
| Hormone-sensitive lipase | *Hsl* | AGGTGGGAATCTCTGCATCACTGT | TGTCCCTGAATAGGCACTGACACA |
| Peroxisome proliferator-activated receptor gamma | *Pparγ* | CAAGAATACCAAAGTGCGATCAA | GAGCAGGGTCTTTTCAGAATAATAAG |
| Lipoprotein lipase | *Lpl* | TGAGAAAGGGCTCTGCCTGA | GGGCATCTGAGAGCGAGTCTT |
| Sterol regulatory element-binding protein 1c | *Srebp-1c* | CTTTCCTGGCTTGTCCTTTGGGA | GCTGGAAGGCAAAGGAACAACTGA |
| Glucose transport type 4 | *Glut4* | GTAACTTCATTGTCGGCATGG | AGCTGAGATCTGGTCAAACG |
| Diacylglycerol *O*-acyltransferase | *Dgat* | CTCTGCCACAGCATTGAGAC | TGCTACGACGAGTTCTTGAG |
| Fatty acid synthase | *Fasn* | CTGGACTCGCTCATGGGTG | CATTTCCTGAAGTTTCCGCAG |
| TATA-box-binding protein | *Tbp* | ACCCTTCACCAATGACTCCTATG | TGACTGGAGCAAATCGCTTGG |
| Clock circadian regulator | *Clock* | ACCACAGCAACAGCAACAAC | GGCTGCTGAACTGAAGGAAG |
| Basic helix-loop-helix ARNT-like protein 1 | *Bmal1* | CCACCTCAGAGCCATTGATACA | GAGCAGGTTTAGTTCCACTTTGTCT |
| Period circadian regulator 1 | *Per1* | TGAAGCAAGACCGGGAGAG | CACACACGCCATCACATCAA |
| Cyclin-dependent kinase inhibitor 2A | *p16* | CCCAACGCCCCGAACT | GCAGAAGAGCTGCTACGTGAA |
| Cyclin-dependent kinase inhibitor 1A | *p21* | GGCAGACCAGCCTGACAGAT | TTCAGGGTTTTCTCTTGCAGAAG |
| Tumour necrosis factor alpha | *Tnf-α* | ATGGCCTCCCTCTCATCAGT | CTTGGTGGTTTGCTACGACG |
| Interleukin 1 beta | *IL-1β* | GCAACTGTTCCTGAACTCAACT | ATCTTTTGGGGTCCGTCAACT |
| Monocyte chemoattractant protein 1 | *Mcp1* | CTTCTGGGCCTGCTGTTCA | GAGTAGCAGCAGGTGAGTGGG |
| Interleukin 6 | *IL-6* | CAAGTCGGAGGCTTAATTACACATG | ATTGCCATTGCACAACTCTTTTCT |

control group, whereas the HFD-TRF group exhibited significantly reduced weight gain relative to both the control and HFD-AL groups (Fig. 2*A*). Specifically at weeks 2, 4, 6 and 7 body weight change in the HFD-TRF group was significantly lower than in the HFD-AL group (Fig. 2*A*). Both average body weight and total fat mass were significantly higher in HFD-AL mice compared to controls (Fig. 2*B* and *C*). TRF partially reversed this HFD-induced increase, though levels remained significantly elevated compared to the control group (Fig. 2*B* and *C*). Lean mass did not differ among groups (Fig. 2*D*) in terms of adipose tissue weight (as a percentage

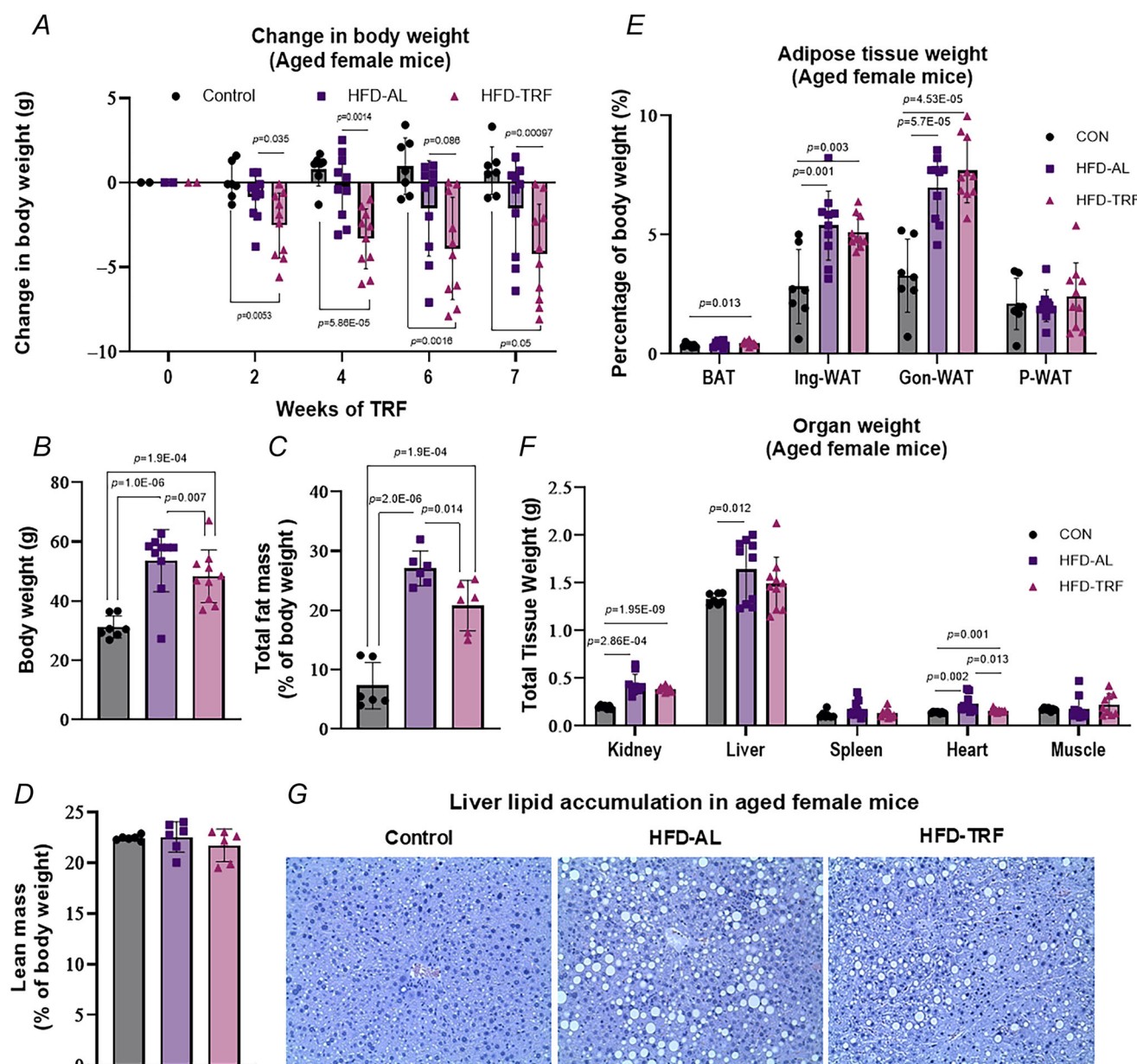

**Figure 2. Effect of TRF on body weight, body composition, tissue weight and liver lipid accumulation in aged female mice with diet-induced obesity**
*A*, change in body weight during the 10 week TRF intervention from aged female mice fed on normal chow (control), high-fat diet *ad libitum* (HFD-AL) and high-fat diet TRF (HFD-TRF). *B*, average body weight at the time of killing after the 10 week TRF intervention. *C*, total fat mass was measured using EchoMRI. *D*, lean mass was measured using EchoMRI. *E*, adipose tissue weights from brown adipose tissue (BAT), inguinal white adipose tissue (Ing-WAT), gonadal white adipose tissue (Gon-WAT) and perirenal white adipose tissue (P-WAT). *F*, weights of non-adipose tissues and organs. *A*, *B*, *E* and *F*, aged female mice – control (*n* = 7), HFD-AL (*n* = 10), HFD-TRF (*n* = 10). *C* and *D*, aged female mice – control (*n* = 6), HFD-AL (*n* = 6), HFD-TRF (*n* = 6). Data are presented as means ± SD. *G*, haematoxylin and eosin (H&E) staining of liver between control, HFD-AL and HFD-TRF groups.

of body weight); the HFD-AL group showed significantly increased inguinal WAT (Ing-WAT) and gonadal WAT (Gon-WAT) compared to controls, with no changes in BAT or perirenal WAt (P-WAT) (Fig. 2*E*). The HFD-TRF group showed significant increases in BAT, Ing-WAT and Gon-WAT compared to controls, with no change in P-WAT (Fig. 2*E*). However there were no significant differences in adipose tissue as a percentage of body weight between the HFD-AL and HFD-TRF groups, except for a trend towards a decrease in Ing-WAT in the HFD-TRF group (Fig. 2*E*). Organ weights were also affected. HFD-AL mice had significantly increased weights of kidney, liver and heart compared to controls (Fig. 2*F*). TRF significantly reduced heart weight and showed a trend towards lower liver and kidney weights compared to HFD-AL mice (Fig. 2*F*). Histological analysis of the liver revealed marked lipid accumulation in the HFD-AL group, indicating significant ectopic fat deposition within the liver (Fig. 2*G*). HFD-TRF mice showed a noticeable reduction in hepatic lipid accumulation compared to HFD-AL mice, although levels remained elevated relative to controls (Fig. 2*G*).

## Effect of TRF on food intake and eating behaviour in aged female mice

Overall food intake was comparable among groups, with slightly increased intake observed in both the HFD-AL and HFD-TRF groups (Fig. 3*A*). During the feeding window the HFD-TRF group consumed significantly more food in both 0–5 h and 5–10 h intervals compared to the control and HFD-AL groups (Fig. 3*B*). In contrast during the inactive phase when HFD-TRF mice had no food access, the control and HFD-AL mice consumed roughly 30% of their daily intake (Fig. 3*B*). Meal frequency analysis revealed a significant increase in the HFD-TRF group compared to the control group during both 0–5 h and 5–10 h periods (the designated feeding windows for TRF) (Fig. 3*C*). Meal frequency in the HFD-TRF group was also significantly higher than the HFD-AL group during the 0–5 h period but not during 5–10 h (Fig. 3*C*). During the inactive period HFD-AL mice exhibited higher meal frequency compared to controls (Fig. 3*C*). No significant differences in meal size were observed among groups, although a downward trend was

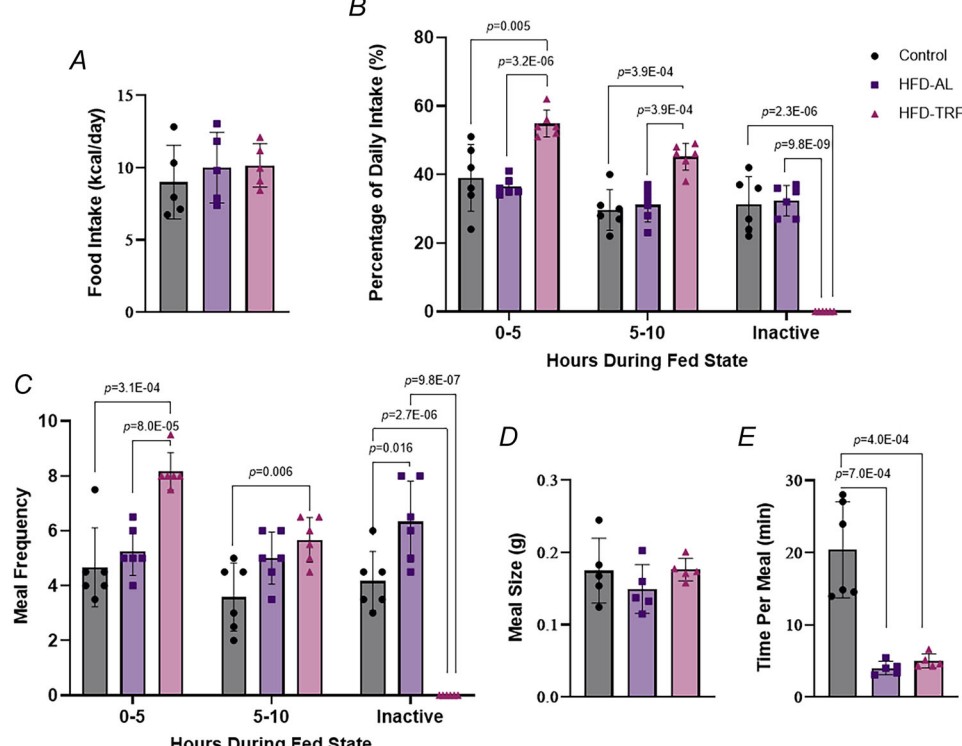

**Figure 3. Effect of TRF on total calorie intake, meal size, meal frequency and meal duration in aged female mice with diet-induced obesity**
*A*, average food intake per day (kcal/day). *B*, percentage of total daily intake broken into three different time periods, the first 5 h of the TRF regimen (0–5), the second 5 h of the TRF regimen (5–10) and the 'inactive time' when TRF mice had no access to food. *C*, frequency of meals broken into three different time periods, the first 5 h of the TRF regimen (0–5), the second 5 h of the TRF regimen (5–10) and the 'inactive time' when TRF mice had no access to food. *D*, average meal size (grams). *E*, meal duration – average time per meal (minutes). All panels: aged female mice – control (*n* = 5–6), HFD-AL (*n* = 5–6), HFD-TRF (*n* = 5–6). Data are presented as means ± SD.

noted in the HFD-AL group (Fig. 3*D*). As shown in Fig. 3*E* meal duration (time per meal) was significantly reduced in both HFD-AL and HFD-TRF groups compared to controls, likely due to the higher caloric density of the HFD.

### Effect of TRF on energy expenditure in aged female mice

Indirect calorimetry analysis showed that both $VO_2$ and $VCO_2$ were significantly reduced during both the light and dark cycles in the HFD-AL and HFD-TRF groups compared to controls (Fig. 4*A* and *B*). However TRF significantly increased $VO_2$ and $VCO_2$ relative to the HFD-AL group, although values remained lower than those in the control group (Fig. 4*A* and *B*). The RER followed a similar pattern during the light cycle (0–12 ZT), with the HFD-TRF group showing an even greater reduction compared to the HFD-AL group. During the dark phase (12–24 ZT) both HFD-AL and HFD-TRF groups exhibited reduced RER compared to controls, with no significant difference between the two groups (Fig. 4*C*). Activity levels during the light cycle significantly decreased in both HFD-AL and HFD-TRF groups of mice compared to controls (Fig. 4*D*). During the dark cycle when mice are typically more active, the HFD-AL group showed a reduction in activity relative to controls, although not significantly. This reduction was partially reversed by TRF (Fig. 4*D*). Heat production increased in both HFD-AL and HFD-TRF groups compared to controls during both the light and dark cycles. Notably heat production in the HFD-TRF group was lower than that in the HFD-AL group, tending back towards the control level (Fig. 4*E*). These results suggest that TRF can partially counteract the HFD-induced reduction in energy expenditure in aged female mice.

### Effect of TRF on white adipose tissue plasticity in aged female mice

To understand how TRF influences adipose tissue plasticity adipocyte size distribution was assessed in Ing-WAT and Gon-WAT. In Ing-WAT HFD feeding led to a marked increase in the 150–200 μm size adipocyte population compared to controls (Fig. 5*A*). TRF significantly reduced the 75–100 μm adipocyte population and increased the 150–200 μm population compared to controls (Fig. 5*A*). Additionally TRF significantly increased the percentage of 25–50 μm adipocytes compared to HFD-AL mice (Fig. 5*A*). Overall the HFD-AL group exhibited a significant increase in average adipocyte diameter compared to both the control and HFD-TRF groups (Fig. 5*B*). Although TRF partially

reversed this enlargement, adipocytes in the HFD-TRF group remained significantly larger than those in the control group (Fig. 5*B*). These morphological changes were confirmed by H&E staining of Ing-WAT sections (Fig. 5*C*).

In contrast more pronounced changes were observed in the Gon-WAT depot. In the HFD-AL group the 75–100 and 100–150 μm adipocyte populations were significantly decreased compared to the control group, whereas the larger adipocyte population (150–200 μm) was increased (Fig. 5*D*). TRF significantly increased the percentage of both 25–50 and 150–200 μm adipocyte populations compared to the control and HFD-AL groups, while reducing the 75–150 μm populations (Fig. 5*D*). In terms of average adipocyte diameter in Gon-WAT both the HFD-AL and HFD-TRF groups showed significant increases compared to the control group (Fig. 5*E*). H&E staining supported these findings, showing enlarged adipocytes in the HFD-AL group relative to the control group, with partial reversal observed in the HFD-TRF group (Fig. 5*C*). These data suggest that TRF promotes a more heterogenous adipocyte population in Gon-WAT, with a mixture of smaller and larger adipocytes, whereas HFD-AL feeding results in a more homogeneous population dominated by larger adipocytes (Fig. 5*D*).

### Effect of TRF on metabolic health in aged female mice

To assess the impact of TRF on metabolic health we evaluated glucose and lipid homeostasis, along with insulin sensitivity. GTT and ITT revealed distinct metabolic responses among the groups. The HFD-AL group exhibited a modest impairment in glucose tolerance compared to the control group, as indicated by elevated blood glucose levels during the GTT (Fig. 6*A*). Insulin sensitivity was significantly reduced in the HFD-AL group, showing higher blood glucose levels during the ITT (Fig. 6*B*). However TRF did not improve glucose tolerance but showed a trend towards enhanced insulin sensitivity in aged female mice on HFD (Fig. 6*B*). Fasting blood glucose levels at the time of killing were significantly elevated in the HFD-AL group compared to the control group (Fig. 6*C*). TRF partially reversed this increase, though not to control levels (Fig. 6*C*). Serum free fatty acids (FFAs) and triglycerides trended upward in the HFD-TRF group relative to the control group (Fig. 6*D* and *E*), but no significant differences were observed between HFD-AL and HFD-TRF groups. Additionally serum $\beta$-hydroxybutyrate levels did not differ significantly among the three experimental groups (Fig. 6*F*).

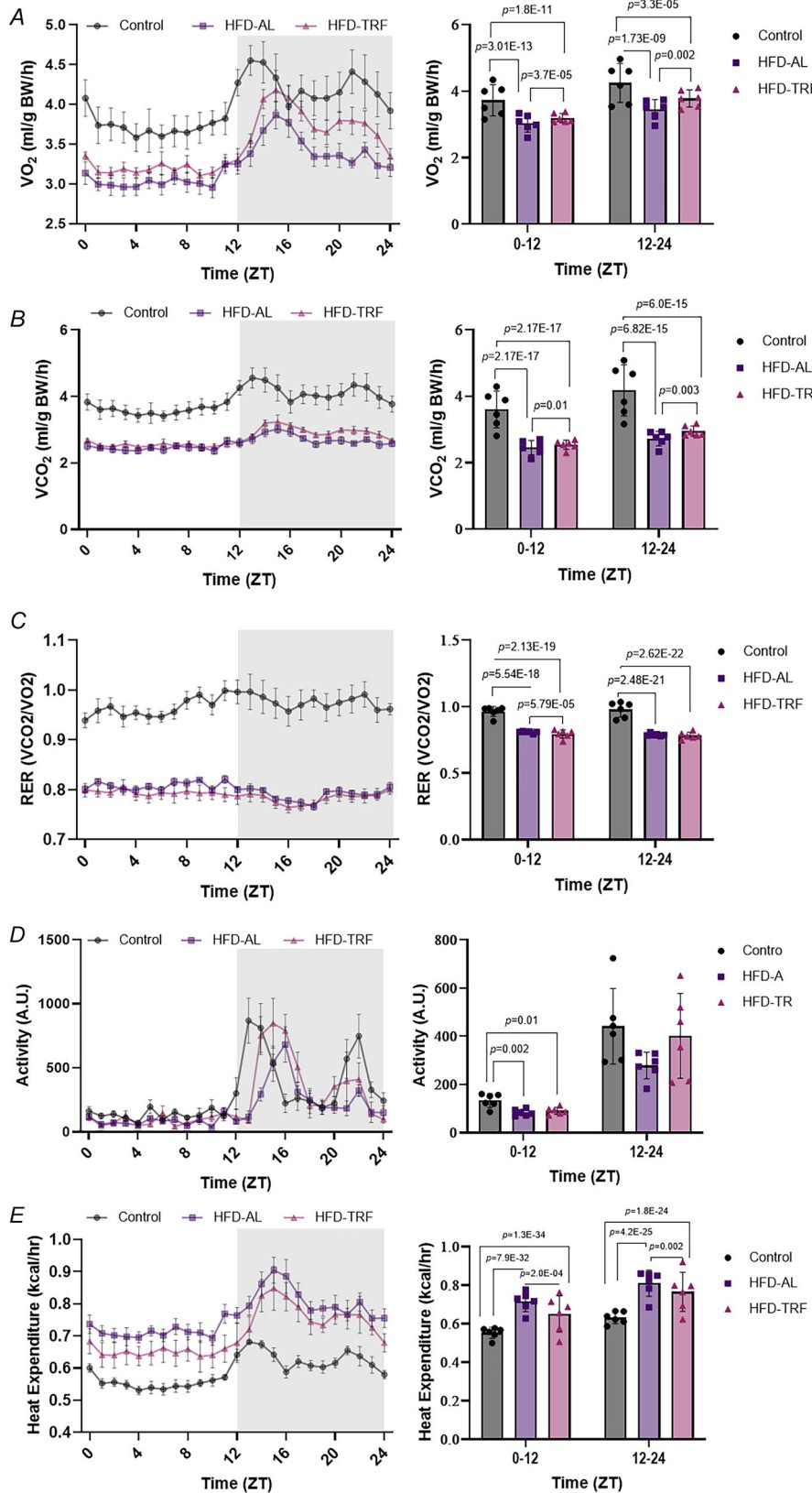

**Figure 4. Indirect calorimetry analysis of aged female mice**

*A*, volume of $O_2$ consumed plotted against time. *B*, volume of $CO_2$ consumed plotted against time. *C*, respiratory exchange ratio ($CO_2$ exhaled/$O_2$ inhaled). *D*, average activity. *E*, heat production (kcal/h). All panels: aged female mice – control (*n* = 6), HFD-AL (*n* = 6), HFD-TRF (*n* = 6). Data are presented as means ± SD.

## Effect of TRF on BAT metabolism in aged female mice

We next examined the effects of TRF on BAT metabolism, focusing on the expression of key genes involved in mitochondrial function, lipid metabolism and thermogenesis. The histological analysis showed increased lipid droplet size in BAT from the HFD-AL group, which was significantly reversed by TRF (Fig. 7*A*). Gene expression analysis showed no significant changes in thermogenic genes such as *Ucp1*, *Cidea* and *Tfam* across the groups (Fig. 7*B*). However genes involved in mitochondrial biogenesis and oxidative metabolism were significantly altered. Specifically *Erra*, *Atp5b* and *Cpt1* were significantly upregulated by HFD-AL, whereas

TRF reversed the expression of *Erra* and *Atp5b* and downregulated *Pgc1a* expression (Fig. 7*D*). HFD-AL also reduced the expression of lipogenic and lipolytic genes, including *Scd1*, *Elovl5* and *Atgl* (Fig. 7*E*). TRF partially or fully restored the expression of these genes, suggesting that TRF can reverse HFD-induced disruption of lipogenesis-lipolysis futile cycle in BAT. Interestingly TRF also restored the expression of circadian rhythm-related genes altered by HFD. *Bmal1* gene expression in BAT was suppressed by HFD relative to controls but was restored to control levels by TRF (Fig. 7*C*). In contrast *Per1* expression was elevated by HFD and reduced back to control levels with TRF (Fig. 7*C*). These findings suggest that TRF can counteract HFD-induced impairments in

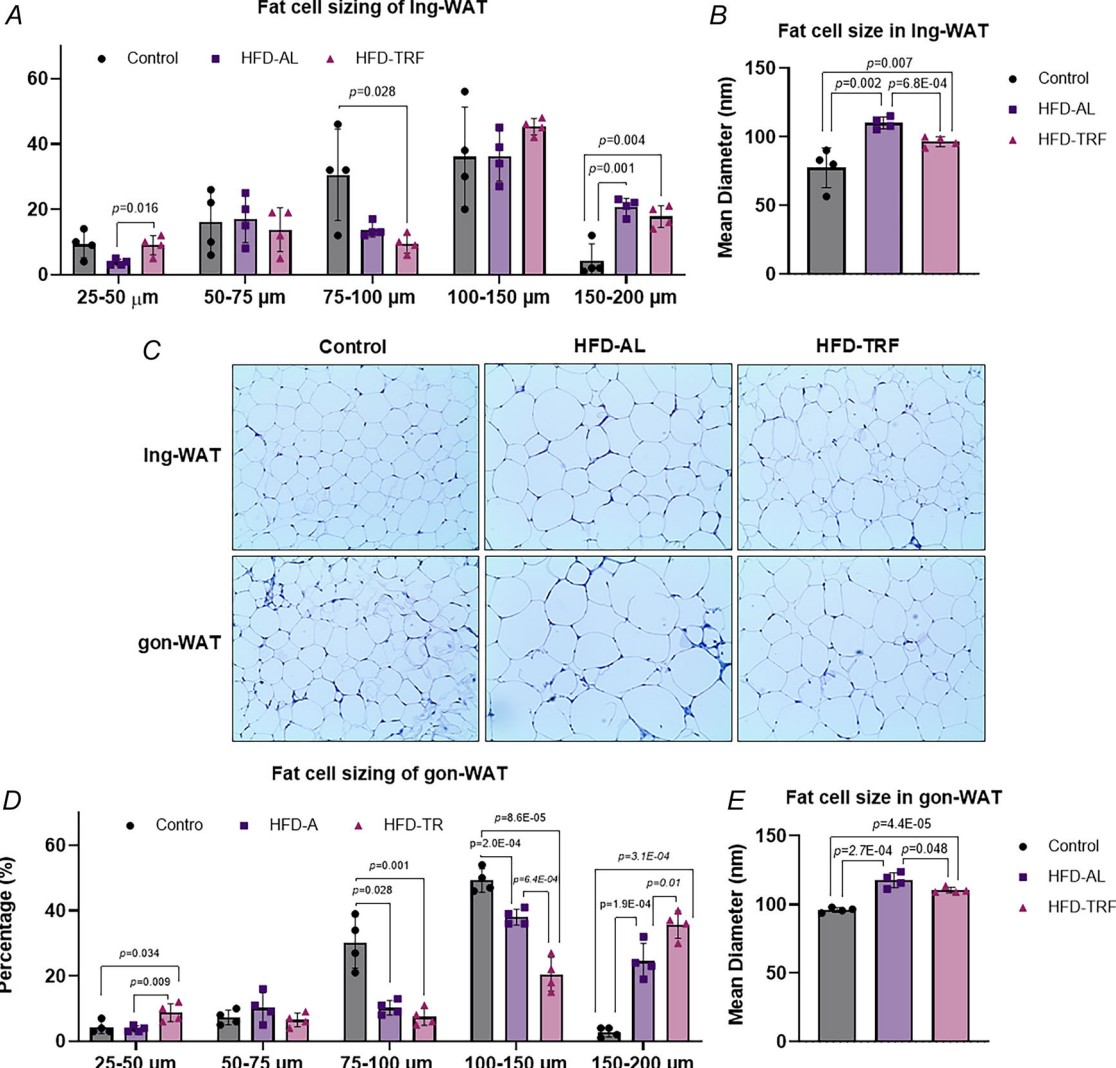

**Figure 5. Effect of TRF on fat cell diameter and sizing distribution in inguinal and gonadal white adipose tissue of aged female mice with diet-induced obesity**
*A, B*, size population distribution and average diameter within the inguinal white adipose tissue. *C, D*, size population distribution and average diameter within the gonadal white adipose tissue. *E*, haematoxylin and eosin (H&E) staining of inguinal and gonadal adipose tissue between control, HFD and TRF groups. *A, B, D*, and *E*: aged female mice – control (*n* = 4), HFD-AL (*n* = 4), HFD-TRF (*n* = 4). Data are presented as means ± SD.

mitochondrial metabolism, lipid cycling and circadian regulation in BAT of aged female mice.

## Effect of TRF on metabolism in gonadal white adipose tissue in aged female mice

We next examined the expression of genes involved in adipogenesis, lipid metabolism, senescence, inflammation and circadian rhythm in Gon-WAT. Expression of adipogenic genes, including *Pparg* and its downstream targets *Lpl* and *Glut4*, was significantly increased by TRF compared to both the control and HFD-AL groups (Fig. 8*A*). *Srebp-1c*, a key transcription factor for lipogenesis, was significantly upregulated by HFD and further increased by TRF compared to the control (Fig. 8*B*). Although *Fasn* expression was suppressed by

HFD, TRF significantly elevated its expression (Fig. 8*B*). Regarding lipolytic genes *Atgl* expression was significantly downregulated by HFD relative to the control (Fig. 8*C*), but TRF restored it towards control levels (Fig. 8*C*). *Hsl* expression did not show significant changes (Fig. 8*C*).

We also assessed circadian rhythm gene expression in Gon-WAT. Both *Clock* and *Bmal1* were significantly decreased in the HDF-AL group compared to the control (Fig. 8*D*), and TRF did not restore their expression (Fig. 8*D*). *Per1* expression showed an increasing trend with HFD, which was significantly reduced by TRF (Fig. 8*D*). For genes associated with senescence, SASP and inflammation *p16* and *p21* (cellular senescence markers) were significantly upregulated by HFD (Fig. 8*E*), and TRF failed to reverse these changes (Fig. 8*E*). The pro-inflammatory cytokines *Tnf*a *and Mcp1* were elevated

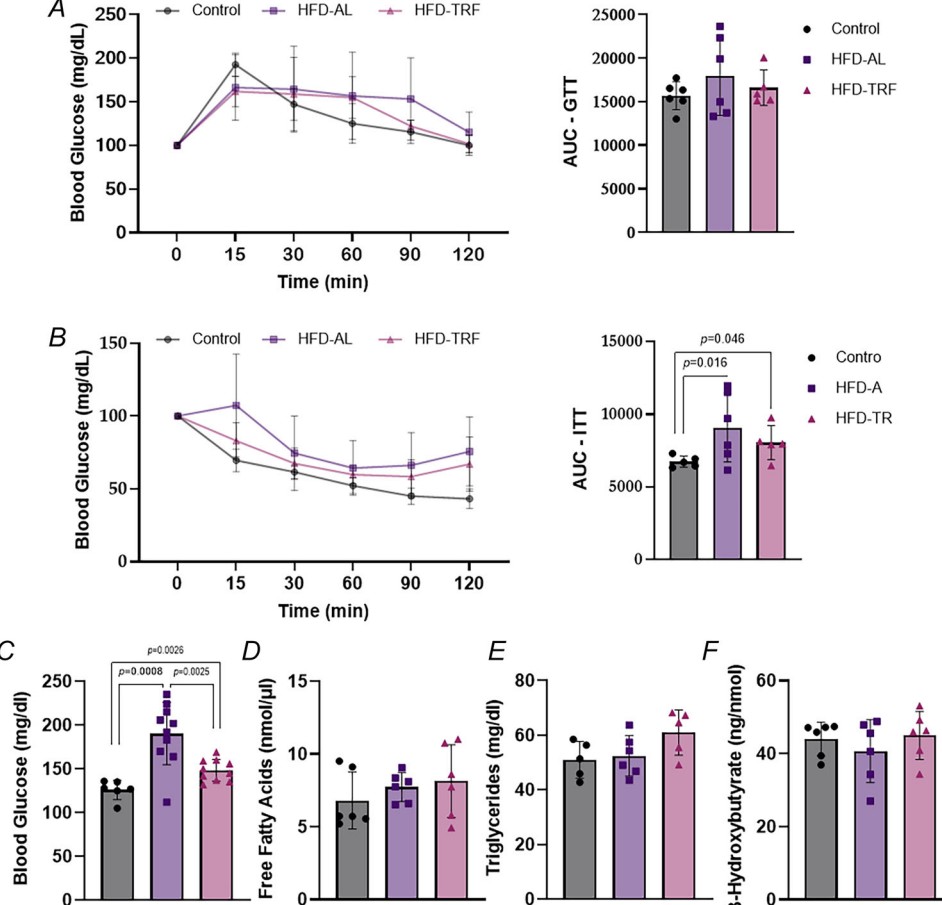

**Figure 6. Effect of TRF on overall metabolic health in aged female mice with diet-induced obesity**
*A*, glucose tolerance at various time points for all experimental groups. Blood glucose levels normalized to 100 mg/dl. *B*, insulin sensitivity test for all experimental groups. Blood glucose levels normalized to 100 mg/dl. *C–F*, measurements of serum glucose and lipid levels in obese aged female mice with 16 h fast. Serum glucose (mg/dl) (*C*), free fatty acid (nmol/µl) (*D*), triglycerides (mg/dl) (*E*) and β-hydroxybutyrate (ng/nmol) (*F*). *A* and *B*, aged female mice: control (*n* = 6), HFD-AL (*n* = 6), HFD-TRF (*n* = 6). *C*, aged female mice: control (*n* = 6), HFD-AL (*n* = 10), HFD-TRF (*n* = 10). *D*, aged female mice: control (*n* = 6), HFD-AL (*n* = 6), HFD-TRF (*n* = 6). *E*, aged female mice: control (*n* = 5), HFD-AL (*n* = 6), HFD-TRF (*n* = 5). *F*, aged female mice: control (*n* = 6), HFD-AL (*n* = 6), HFD-TRF (*n* = 6). Data are presented as means ± SD.

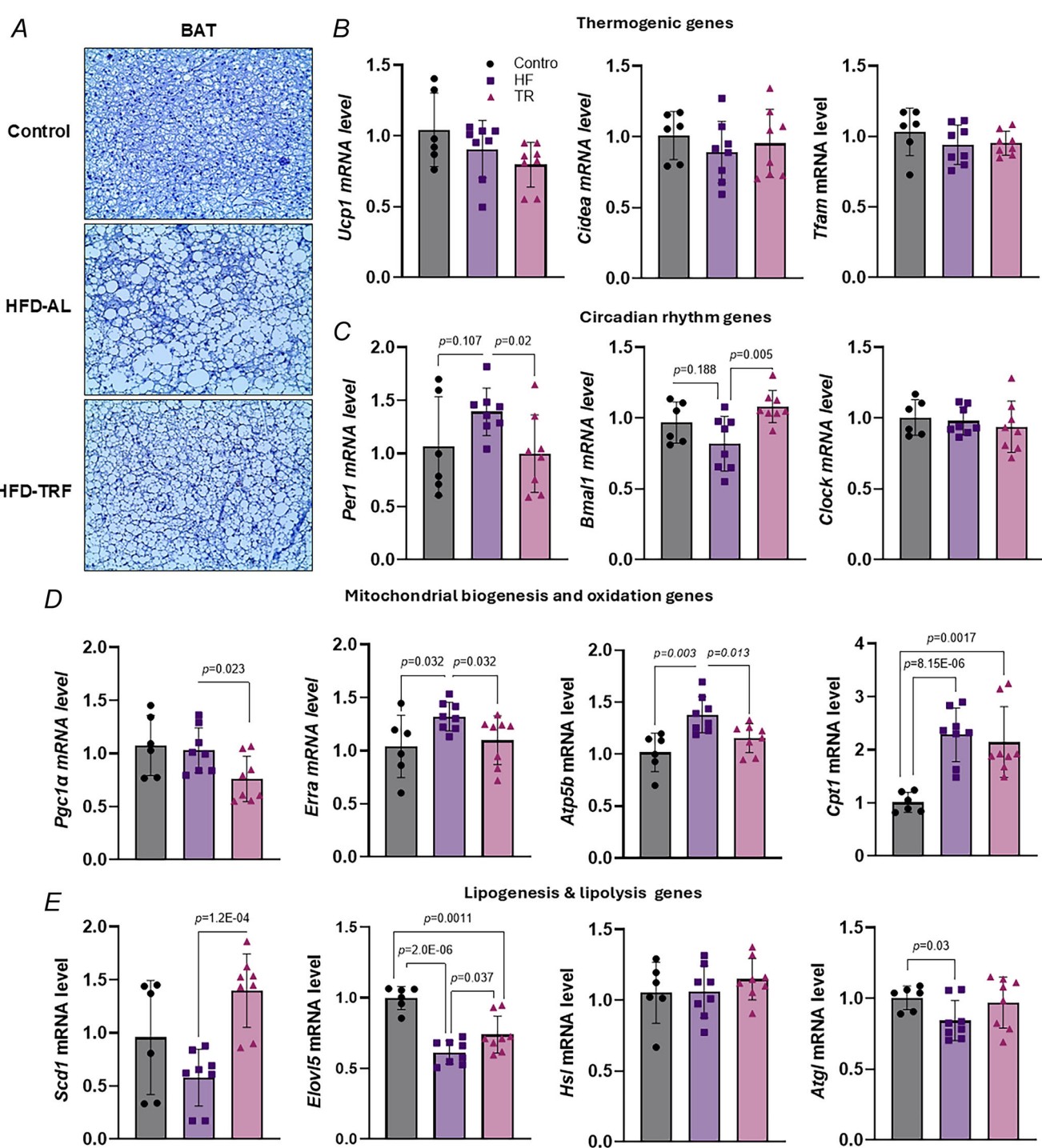

**Figure 7. Effect of TRF on brown adipose tissue metabolism in aged female mice with diet-induced obesity**
*A*, haematoxylin and eosin (H&E) staining of brown adipose tissue between control, HFD-AL and HFD-TRF groups. *B–P*, mRNA expression of genes involved in thermogenesis, mitochondrial biogenesis, fatty acid oxidation, lipogenesis, lipolysis and circadian rhythm in brown adipose tissue of control, HFD-AL and HFD-TRF mice. *B–E*, aged female mice – control (*n* = 6), HFD-AL (*n* = 8), HFD-TRF (*n* = 8). Data are presented as means ± SD.

in the HFD-AL group (Fig. 8*E*), but TRF partially mitigated these increases (Fig. 8*E*). Overall these findings indicate that TRF can modulate gene expression in Gon-WAT by upregulating adipogenic genes and partially reversing mitigating HFD-induced changes in lipogenesis, lipolysis and inflammation. However TRF had limited effects on senescence markers and circadian rhythm genes, highlighting the complex, depot-specific interactions between diet, metabolic health and circadian regulation.

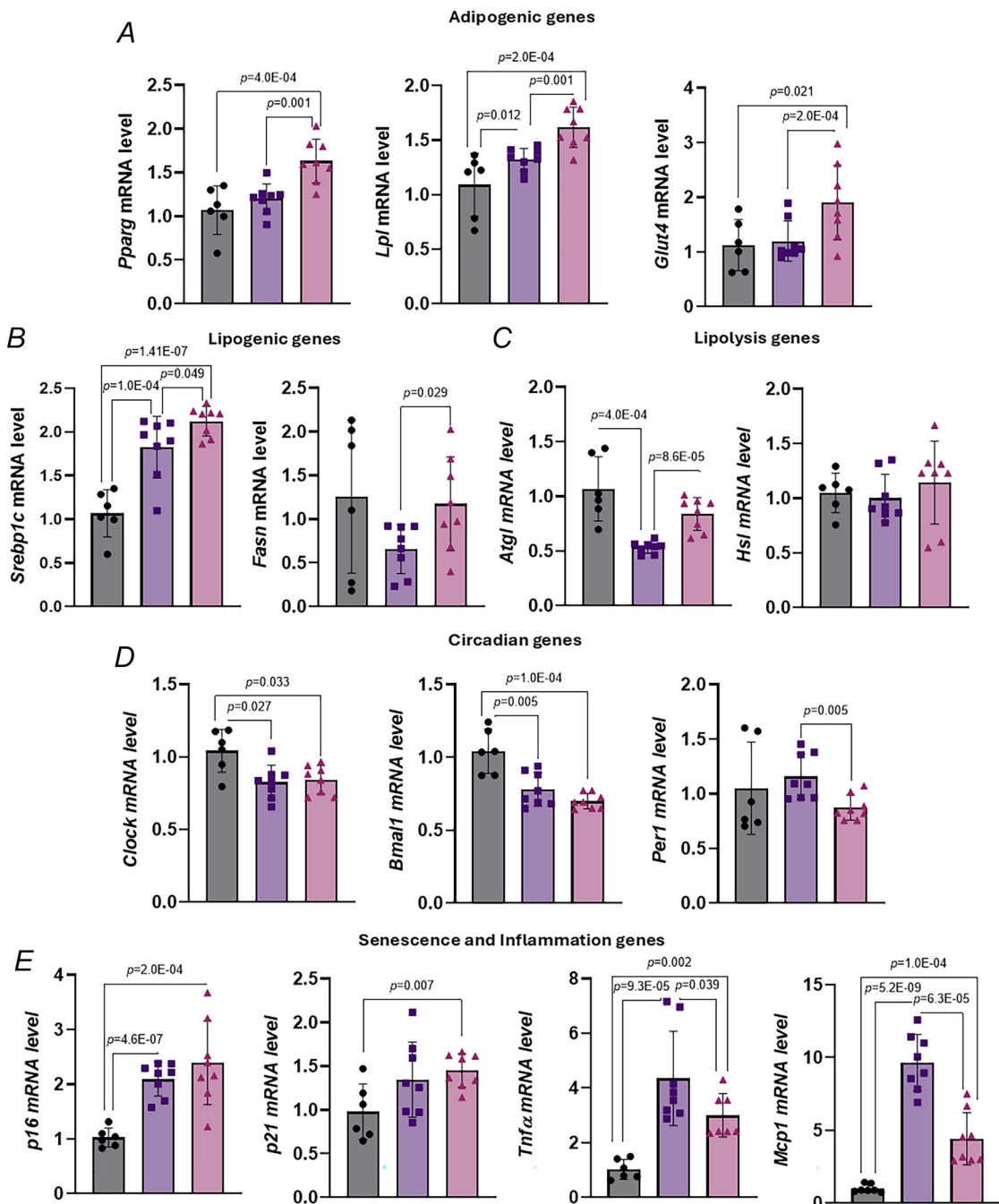

**Figure 8. Effect of TRF on gonadal white adipose tissue metabolism in aged female mice with diet-induced obesity**

*A–D*, mRNA expression of genes involved in adipogenesis/lipogenesis lipolysis, and circadian rhythm in gonadal white adipose tissue of control, HFD-AL and HFD-TRF mice. *E*, mRNA expression of genes involved in senescence and inflammation in gonadal white adipose tissue of control, HFD-AL and HFD-TRF mice. All panels: aged female mice – control (*n* = 6), HFD-AL (*n* = 8), HFD-TRF (*n* = 8). Data are presented as means ± SD.

## Effect of TRF on metabolism in inguinal white adipose tissue in aged female mice

To explore how TRF affects metabolic adaptations in Ing-WAT we analysed the expression of genes involved in adipogenesis, lipid metabolism, mitochondrial function, senescence, inflammation and circadian rhythm. *Pparg* and its target gene *Glut4* was significantly upregulated by HFD, whereas *Lpl* showed no significant change (Fig. 9*A*). TRF significantly increased *Lpl* expression beyond control levels and showed a trend towards increased *Glut4* (Fig. 9*A*), indicating a partial reversal of HFD-induced

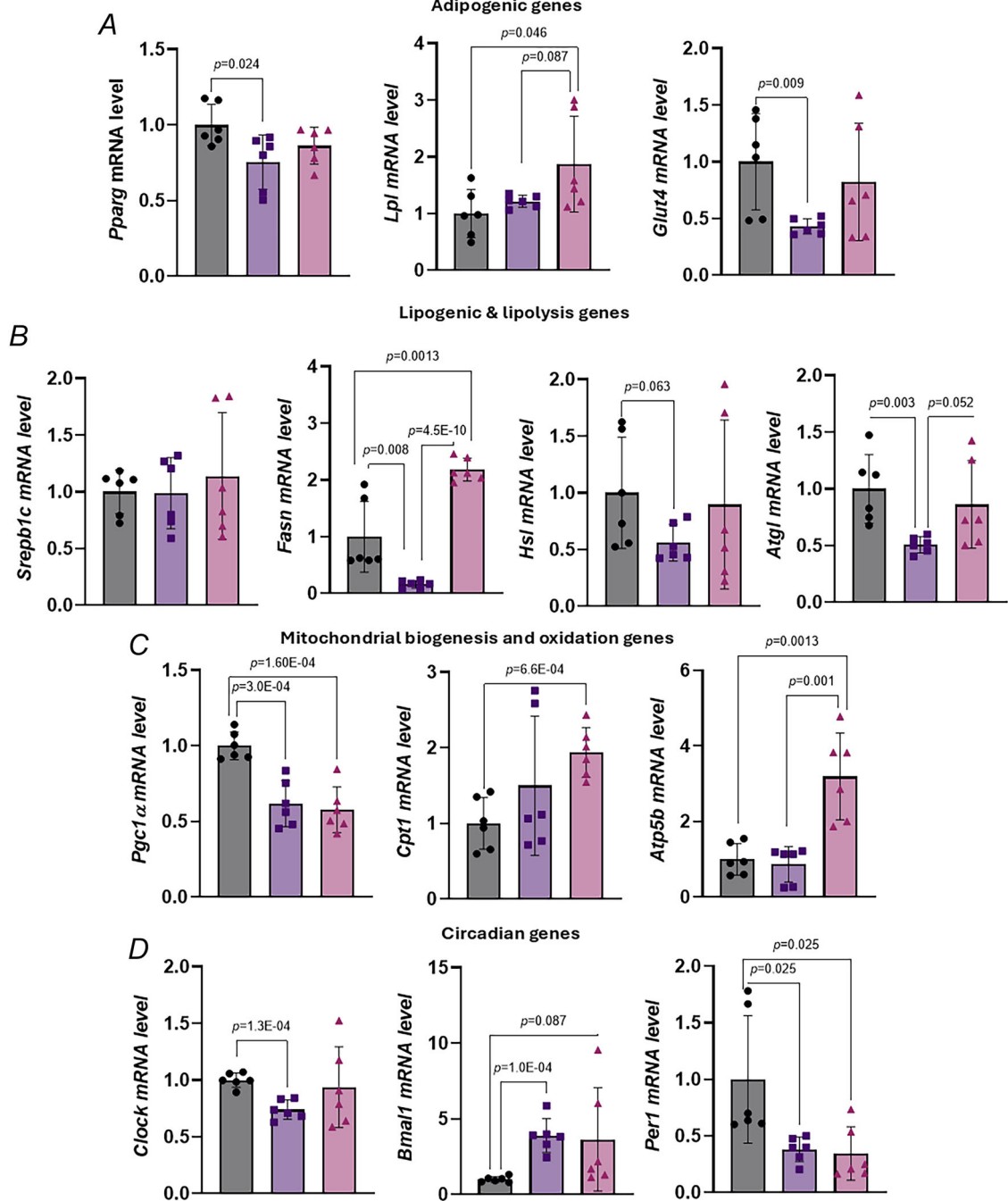

**Figure 9. Effect of TRF on inguinal white adipose tissue metabolism in aged female mice with diet-induced obesity**

*A–D*, mRNA expression of genes involved in adipogenesis/lipogenesis lipolysis, mitochondrial biogenesis and oxidation, and circadian rhythm in inguinal white adipose tissue of control, HFD-AL and HFD-TRF mice. All panels: aged female mice – control (*n* = 6), HFD-AL (*n* = 6), HFD-TRF (*n* = 6). Data are presented as means ± SD.

adipogenic impairment. Lipogenic gene *Fasn* and lipolytic genes *Atgl* and *Hsl* were significantly downregulated by HFD (Fig. 9*B*). TRF reversed the suppression of *Fasn* and *Atgl* but had no significant effect on *Hsl* expression (Fig. 9*B*). For mitochondrial function *Atp5b* and *Cpt1* were significantly upregulated by TRF compared to both the control and HFD-AL groups (Fig. 9*C*), suggesting enhanced mitochondrial oxidation and ATP production in Ing-WAT. However TRF had no reversal effect on the HFD-induced downregulation of *Pgc1a* gene (Fig. 9*C*).

The assessment of circadian rhythm genes revealed that the expression of *Clock* and *Per1* genes was decreased by HFD (Fig. 9*D*), whereas *Bmal1* was significantly upregulated (Fig. 9*D*). TRF did not significantly alter these expression patterns (Fig. 9*D*). Collectively these findings suggest that TRF improves lipogenesis-lipolysis cycling and enhances mitochondrial oxidative capacity in Ing-WAT, although it has limited effects on circadian gene expression in aged female mice.

## Effect of TRF on weight gain and adiposity in middle-aged female mice

To determine whether ageing impacts the effectiveness of TRF we also investigated the effects of TRF in middle-aged female mice. As shown in Fig. 10 the HFD-AL group exhibited significant weight gain in middle-aged female mice (Fig. 10*A* and *B*), whereas body weight in aged HFD-fed females remained stable throughout the TRF regimen (Fig. 2*A*). Consequently TRF induced more pronounced weight changes in middle-aged female mice (Fig. 10*A*) compared to their HFD-AL counterparts in aged female mice (Fig. 2*A*). Furthermore TRF led to a greater reduction in adipose tissue mass, with significant decreases observed in Ing-WAT, Gon-WAT and perirenal WAT (Fig. 10*C*). However in contrast to aged female mice TRF did not significantly affect organ weights in middle-aged females (Fig. 10*D*). These differences may reflect age-related variations in metabolic adaptations, which could influence responsiveness to dietary interventions such as HFD and TRF.

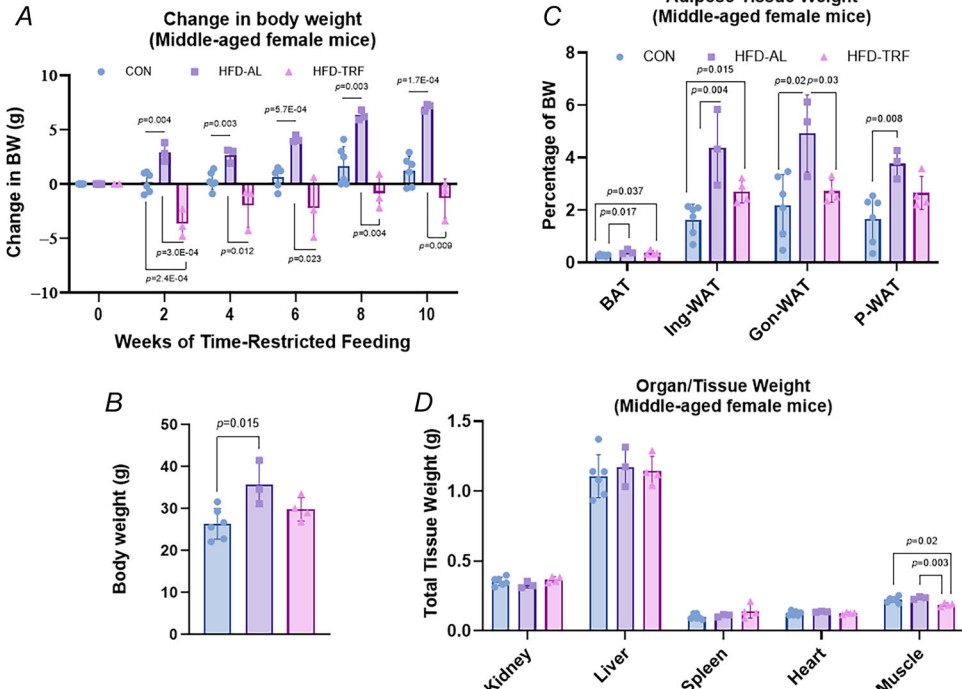

**Figure 10. Effect of TRF on body weight and tissue weights in middle-aged female mice with diet-induced obesity**
*A*, change in body weight during the 10 weeks of TRF for middle-aged female mice on normal chow (control), high-fat diet *ad libitum* (HFD-AL) and high-fat diet TRF (HFD-TRF). *B*, average body weight at the time of killing after the 10 week TRF implementation. *C*, adipose tissue weight from brown adipose tissue (BAT), inguinal white adipose tissue (Ing-WAT), gonadal white adipose tissue (Gon-WAT) and perirenal white adipose tissue (P-WAT). *A*–*D*, middle-aged female mice – control (*n* = 6), HFD-AL (*n* = 3), HFD-TRF (*n* = 3–4). Data are presented as means ± SD.

## Discussion

In this study we investigate the impact of TRF on metabolic adaptations related to body weight, meal patterns, energy expenditure, adipose tissue plasticity and metabolism in aged female mice. Consistent with previous studies that TRF reduces weight gain in younger male mice (Chaix et al., 2019; Hatori et al., 2012; Lee et al., 2021) our results demonstrate that TRF is also effective for promoting weight loss in aged female mice fed a HFD. TRF partially reversed the HFD-induced increases in body weight and total fat mass, although both remained significantly higher than those observed in the control group. The observed reductions in fat mass between the TRF and HFD groups may be partially explained by TRF-induced decreases in the weight of heart, liver and kidneys, suggesting a reduction in ectopic fat deposition, as evidenced by lower hepatic lipid accumulation. The decrease in heart weight observed with TRF may indicate a cardioprotective effect in the context of HFD, which is particularly relevant given the increased risk of cardiovascular diseases associated with obesity and HFD (Powell-Wiley et al., 2021). When evaluating the effect of age on TRF outcomes, middle-aged female mice showed greater reductions across all three adipose depots compared to aged females. Notably lean mass remained unchanged across groups, indicating that the weight loss observed was primarily due to reductions in fat mass rather than lean tissue. This preservation of lean mass is particularly important in ageing populations, as muscle mass plays a critical role in metabolic health and physical function (Kim & Kim, 2020). When tissue weights were normalized to body weight, TRF failed to fully prevent HFD-induced increases in Ing-WAT and Gon-WAT. However the increase in BAT with TRF is notable, as BAT is associated with thermogenesis and increased energy expenditure (Townsend & Tseng, 2014). Importantly the increase in BAT is not due to greater fat accumulation; in fact BAT from the TRF mice shows reduced fat deposition compared to the HFD group. These findings suggest that TRF could promote beneficial metabolic adaptations in aged females, potentially enhancing energy expenditure and offering a therapeutic strategy to combat diet-induced metabolic dysfunction.

Next we explored the effects of TRF on energy expenditure using indirect calorimetry. Although TRF does not fully restore metabolic rates to control levels, it significantly increased both $VO_2$ and $VCO_2$ relative to the HFD-AL group, indicating that TRF partially counteracts some metabolic slowdowns associated with HFD consumption. RER, defined as the ratio of $VCO_2$ to $VO_2$ and indicative of substrate utilization, followed a similar pattern. During the light cycle RER values in the TRF group were further reduced compared to the HFD group, suggesting a greater reliance on fat oxidation. This trend was not observed during the dark cycle, indicating that TRF may promote a metabolic shift towards fat utilization specifically during the fasting state. Activity levels were reduced in both the HFD and TRF groups during the light cycle; however TRF partially reversed the HFD-induced reduction in activity during the dark cycle. This indicates that TRF helps sustain higher physical activity levels even under metabolic stress from a HFD. Increased activity may be associated with heightened feeding behaviour, as evidenced by a rise in meal frequency in the TRF group. TRF also reduced heat production relative to the HFD group, bringing it closer to control levels. This suggests improved metabolic efficiency. In contrast the increased heat production observed in the HFD group may reflect a compensatory thermogenic response to excessive caloric intake. Supporting this HFD-fed mice showed an increased expression of *Cpt1* and *Atp5b* in BAT, indicating enhanced mitochondrial activity and ATP production, both key components of thermogenesis and energy expenditure (Calderon-Dominguez et al., 2016). Previous studies have demonstrated that TRF effectively reduces body weight and improves metabolic parameters such as insulin sensitivity, glucose levels, cholesterol and inflammation in young male mice (Hatori et al., 2012; Sherman et al., 2012). In contrast the attenuated response to TRF observed in aged female mice in this study may reflect age-related metabolic inflexibility or altered hormonal signalling that affects adaptation to dietary interventions.

Our analysis of meal patterns and overall food intake in aged female mice revealed significant alterations driven by both HFD and TRF. The time per meal significantly decreased in both HFD and TRF groups compared to controls, likely due to the higher caloric density of the HFD. Total food intake and average meal size remained similar across all groups, indicating that TRF does not lead to a reduction in overall caloric intake. Mice in the HFD-TRF group exhibited a higher meal frequency during the 0–5 h period compared to HFD-AL mice, reflecting behavioural adaptation to the restricted feeding window. In contrast HFD-AL mice showed increased meal frequency during the inactive (light) period relative to controls, indicating that HFD disrupts normal feeding rhythms. This shift towards increased feeding during the rest phase may contribute to adverse metabolic outcomes, as misaligned feeding has been shown to disrupt metabolic homoeostasis and circadian rhythm (Opperhuizen et al., 2016; Pickel & Sung, 2020). HFD-TRF mice consumed a larger proportion of their daily intake during the designated feeding window relative to both control and HFD-AL mice, consistent with the TRF protocol. Notably control and HFD-AL mice consumed approximately 30% of their daily calories during the inactive period. In typical conditions mice fed a normal chow diet *ad libitum* consume about 20% of their

total intake during this phase (Chaix & Zarrinpar, 2015). However HFD feeding often results in more irregular intake patterns, with 30%–40% of calories consumed during the inactive period (Hatori et al., 2012; Kohsaka et al., 2007). Although a previous study in middle-aged (3 months old) mice reported that TRF led to reduced overall food intake and body weight stabilization (Hepler et al., 2022), our findings in aged female mice indicate that TRF primarily alters meal timing and frequency rather than reducing total caloric consumption. This supports previous studies demonstrating that the metabolic benefits of TRF may derive more from aligning feeding with circadian rhythms than from calorie restriction alone (Chaix et al., 2014; Hatori et al., 2012).

In terms of adipose tissue plasticity our findings revealed significant changes in adipocyte size distribution within Ing-WAT and Gon-WAT. TRF reduced the mean adipocyte diameter in both depots, suggesting that it mitigates HFD-induced adipocyte hypertrophy and associated dysfunction. Smaller adipocytes are typically linked to improved insulin sensitivity and better metabolic regulation (Bernstein et al., 1975; Haller et al., 1979; Krotkiewski et al., 1983). However the continued presence of larger adipocytes in TRF-treated mice indicates that TRF may be insufficient to fully restore healthy adipose tissue function in aged populations.

Both HFD-AL and HFD-TRF groups exhibited comparable impairments in glucose tolerance, suggesting that TRF failed to prevent HFD-induced glucose intolerance in aged female mice. Although the difference did not reach statistical significance, TRF partially reversed HFD-induced insulin resistance, suggesting some improvement in insulin sensitivity. Furthermore fasting blood glucose levels after a 16 h fast were significantly lower in the HFD-TRF group compared to HFD-AL mice, indicating that TRF partially ameliorates chronic hyperglycaemia associated with long-term HFD exposure. Despite elevated levels of FFAs and triglycerides being commonly associated with enhanced lipolysis and a shift towards fat utilization, aligning with changes in adipose tissue morphology and adipocyte size distribution (Duncan et al., 2007; Grabner et al., 2021), our study found no significant changes in circulating FFAs or triglycerides among groups. Although TRF has been shown to provide robust protection against HFD-induced metabolic disturbance in middle-aged mice, where it effectively prevents or reverses metabolic impairments (Chaix et al., 2019; Hatori et al., 2012; Sutton et al., 2018), our findings suggest that in aged female mice TRF alone is insufficient to fully restore metabolic health. This highlights the potential for age-related decline in the adaptive capacity to dietary interventions

Next we examined the expression of genes involved in adipose tissue metabolism to assess the impact of TRF on metabolic adaptations in aged female mice.

Interestingly the expression levels of key thermogenic genes, *Ucp1*, *Cidea* and *Tfam*, remained unchanged across experimental groups. This suggests that TRF does not modulate lipid metabolism in BAT through UCP1-mediated thermogenesis in aged female mice. HFD feeding is known to upregulate mitochondrial oxidative genes and overload mitochondrial function (Koves et al., 2008; Lai et al., 2020), resulting in incomplete oxidation of long-chain fatty acids and mitochondrial dysfunction. In line with this we observed significant upregulation of mitochondrial oxidation-related genes *Errα*, *Atp5b* and *Cpt1* in HFD-fed mice. Notably TRF attenuated the HFD-induced expression of *Errα*, *Pgc-1α*, *Atp5b* and *Cpt1*, suggesting that TRF improves mitochondrial oxidative function and normalizes ATP production in BAT (Gaillard et al., 2006). This may underlie the increased energy expenditure observed in TRF-treated aged mice, as indicated by elevated $VO_2$ and $VCO_2$. However the UCP1-dependent thermogenic pathway appears unaffected by TRF. We next explored whether TRF increases energy expenditure through UCP1-independnet mechanisms, such as ATP-consuming futile cycles (Brownstein et al., 2022). One such mechanism is lipid cycling, involving simultaneous lipid synthesis (lipogenesis) and breakdown (lipolysis), which expends energy without net fat storage. In our study HFD significantly suppressed the expression of both lipogenic and lipolytic genes, where TRF restored their expression in BAT, Ing-WAT and Gon-WAT. This indicates that TRF reinstates HFD-disrupted lipid cycling. Additionally TRF significantly increases adipogenesis across all three fat depots. Collectively these results suggest that TRF can efficiently restore or prevent HFD-induced disruption of lipid synthesis-breakdown cycling in adipose tissue, thereby contributing to increased energy expenditure in aged female mice. TRF also significantly upregulated *Srebp-1c* and *Pparg*, kry transcription factors for lipogenesis and adipocyte differentiation, specifically in Gon-WAT. This may reflect an enhanced lipogenic response aimed at managing lipid overflow from the HFD (Unger, 2003), consistent with the more pronounced fat cell size shifts observed in this depot, including an increased proportion of smaller adipocytes relative to Ing-WAT (Medina-Gomez et al., 2007).

Adipose tissue metabolism is under strong circadian regulation, and HFD-induced metabolic stress has been shown to impair circadian clock function, potentially through inflammation, altered nutrient signalling and oxidative stress (Haspel et al., 2014; Kim et al., 2018; Sherman et al., 2012). The core circadian genes *Clock*, *Bmal1* and *Per1* co-ordinate various metabolic processes (Manoogian & Panda, 2017). In our study the expression of these circadian genes was significantly altered by both HFD and TRF, with distinct patterns emerging across fat depots. Notably TRF restored circadian gene expression

in BAT but not in WATs, indicating a depot-specific effect. The underlying mechanisms driving this selective response remain to be elucidated and warrant further investigation.

## Conclusion

We have demonstrated that restricting feeding during the light cycle can restore circadian metabolic rhythms and enhance mitochondrial oxidative function in BAT, thereby increasing energy expenditure. This feeding pattern also activates the futile lipid cycle (lipolysis-lipogenesis) in adipose tissue, likely driven by elevated FA release during the light cycle, further contributing to increased energy expenditure. Additionally light cycle feeding restrictions increase meal frequency (i.e. feeding activity) during the dark cycle, suggesting enhanced behavioural adaptation to TRF. Collectively TRF induces beneficial metabolic adaptations in adipose tissue that promote energy expenditure and improve overall metabolic health. These adaptations include reductions in HFD-induced weight gain, ectopic fat accumulation, fasting blood glucose levels and adipose tissue inflammation. Notably these metabolic effects of TRF are depot-specific, varying across different fat depots. In conclusion our findings suggest that TRF is a promising dietary approach for improving metabolic health in HFD-fed older female mice. Given that Western diets are typically higher in fat and strongly associated with rising rates of obesity and metabolic disorders, this model provides important insight into how altered feeding patterns can counteract diet-induced metabolic dysfunction. However one limitation of this study is the lack of comparison between the beneficial effects of TRF in young *vs.* aged female mice. Additionally we did not examine the impact of TRF in postmenopausal females under normal dietary conditions.

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

## Additional information

### Data availability statement

The datasets generated and analysed during the current study are available in Supporting Information. Additional data and analysis are available from the corresponding author on reasonable request.

### Competing interests

The authors declare that they have no competing interests.

### Author contributions

T.B. designed the study, conducted the experiments, analysed the data, prepared the figures, wrote the manuscript and acquired funding for the project. H.S. conducted the experiments. X.C. contributed to the conceptualization of the study, provided supervision, acquired funding for the project, edited the manuscript and approved the final version. All authors have approved the final version of the manuscript and agreed to be accountable for all aspects of the work. All people designated as authors qualify for authorship, and all those who qualify for authorship are listed.

### Funding

This research was supported by Healthy Foods Healthy Lives Graduate Student Research Grant (HFHL 2022) awarded from Healthy Foods, Healthy Lives Institute at the University of Minnesota and NIDDK Grant (R01 DK123042).

### Acknowledgements

Portions of this work were conducted in the Department of Integrative Biology and Physiology at the University of Minnesota. Portions of this work were conducted in the Minnesota Nano Centre, which is supported by the National Science Foundation through the National Nanotechnology Coordinated Infrastructure (NNCI) under award number ECCS-2025124. We would also like to thank Drs Alessandro Bartolomucci and Maria Razzoli at the Integrative Biology and Physiology Phenotyping Core, University of Minnesota-Twin Cities, for their expertise and assistance with the analysis of meal pattern and indirect calorimetry.

### Keywords

adipose tissue, high-fat diet, metabolism, nutrition, obesity, time-restricted feeding

### Supporting information

Additional supporting information can be found online in the Supporting Information section at the end of the HTML view of the article. Supporting information files available:

**Peer Review History**

