## [Peer Review History · The Journal of Physiology]

Impact of time-restricted feeding on metabolic health and adipose tissue metabolism in aged female mice with high fat diet-induced obesity

Theresa Bushman, Hongming Su, and Xiaoli Chen
DOI: 10.1113/JP289464

Corresponding author(s): Xiaoli Chen (xlchen@umn.edu)

The following individual(s) involved in review of this submission have agreed to reveal their identity: Sophie Seward (Referee #1); Lamy'a Majdi Dawud (Referee #2)

Review Timeline:

Submission Date:	11-Jun-2025
Editorial Decision:	26-Aug-2025
Revision Received:	01-Oct-2025
Editorial Decision:	20-Nov-2025
Revision Received:	11-Dec-2025
Accepted:	17-Dec-2025

Senior Editor: Karyn Hamilton

Reviewing Editor: Josiane Broussard

Transaction Report:

Dear Dr Chen,

Re: JP-RP-2025-289464 **"Impact of time-restricted feeding on metabolic health and adipose tissue metabolism in aged female mice with high fat diet-induced obesity"** by Xiaoli Chen, Theresa Bushman, and Hongming Su

Thank you for submitting your manuscript to The Journal of Physiology. It has been assessed by a Reviewing Editor and by 2 expert referees and we are pleased to tell you that it is potentially acceptable for publication following satisfactory major revision.

REVISION CHECKLIST:

Please upload two versions of your manuscript text: one with all relevant changes highlighted and one clean version with no

changes tracked. The manuscript file should include all tables and figure legends, but each figure/graph should be uploaded as separate, high-resolution files.

We look forward to receiving your revised submission.

Yours sincerely,

Karyn Hamilton
Senior Editor
The Journal of Physiology

REQUIRED ITEMS

- 1) - Include a Key Points list in the article itself, before the Abstract.
- 2) - Author photo and profile. First or joint first authors are asked to provide a short biography (no more than 100 words for one author or 150 words in total for joint first authors) and a portrait photograph. These should be uploaded and clearly labelled together in a Word document with the revised version of the manuscript. See Information for Authors for further details.
- 3) - You must start the Methods section with a paragraph headed Ethical approval (https://jp.msubmit.net/cgi-bin/main.plex?form_type=display_requirements#methods).

Research must comply with The Journal's policies regarding animal experiments (<https://physoc.onlinelibrary.wiley.com/hub/animal-experiments>) and adherence to these policies must be stated in the manuscript.

Authors should confirm in their Methods section that their experiments were carried out according to the guidelines laid down by their institution's animal welfare committee, including an ethics approval reference number. The Methods section must contain a statement about access to food, water and housing, details of the anaesthetic regime: anaesthetic used, dose and route of administration, and method of killing the experimental animals.

- 4) - Your manuscript must include a complete Additional Information section, including competing interests; funding; author contributions and acknowledgements.
- 5) - Please upload separate high-quality figure files via the submission form.
- 6) - Please ensure that any tables are editable and in Word format, and wherever possible, embedded in the article file itself.
- 7) - Please ensure that the Article File you upload is a Word file.
- 8) - Papers must comply with the Statistics Policy: https://jp.msubmit.net/cgi-bin/main.plex?form_type=display_requirements#statistics.

In summary:

- If n {less than or equal to} 30, all data points must be plotted in the figure in a way that reveals their range and distribution. A bar graph with data points overlaid, a box and whisker plot or a violin plot (preferably with data points included) are acceptable formats.

- If $n > 30$, then the entire raw dataset must be made available either as supporting information, or hosted on a not-for-profit repository, e.g. FigShare, with access details provided in the manuscript.
- 'n' clearly defined (e.g. x cells from y slices in z animals) in the Methods. Authors should be mindful of pseudoreplication.
- All relevant 'n' values must be clearly stated in the main text, figures and tables.
- The most appropriate summary statistic (e.g. mean or median and standard deviation) must be used. Standard Error of the Mean (SEM) alone is not permitted.
- Exact p values must be stated. Authors must not use 'greater than' or 'less than'. Exact p values must be stated to three significant figures even when 'no statistical significance' is claimed.

9) - A Data Availability Statement is required for all papers reporting original data. This must be in the Additional Information section of the manuscript itself. It must have the paragraph heading 'Data Availability Statement'. All data supporting the results in the paper must be either: in the paper itself; uploaded as Supporting Information for Online Publication; or archived in an appropriate public repository. The statement needs to describe the availability or the absence of shared data. Authors must include in their statement: a link to the repository they have used, or a statement that it is available as Supporting Information; reference the data in the appropriate sections(s) of their manuscript; and cite the data they have shared in the References section. Whenever possible, the scripts and other artefacts used to generate the analyses presented in the paper should also be publicly archived. If sharing data compromises ethical standards or legal requirements then authors are not expected to share it, but must note this in their statement. For more information, see our Statistics Policy.

10) - Please include an Abstract Figure file, as well as the Figure Legend text within the main article file. The Abstract Figure is a piece of artwork designed to give readers an immediate understanding of the research and should summarise the main conclusions. If possible, the image should be easily 'readable' from left to right or top to bottom. It should show the physiological relevance of the manuscript so readers can assess the importance and content of its findings. Abstract Figures should not merely recapitulate other figures in the manuscript. Please try to keep the diagram as simple as possible and without superfluous information that may distract from the main conclusion(s). Abstract Figures must be provided by authors no later than the revised manuscript stage and should be uploaded as a separate file during online submission labelled as File Type 'Abstract Figure'. Please also ensure that you include the figure legend in the main article file. All Abstract Figures should be created using BioRender. Authors should use The Journal's premium BioRender account to export high-resolution images. Details on how to use and access the premium account are included as part of this email.

11)- Please ensure that all figures and tables have a title and legend, and that they have been cited within the main article text. The Figure legends have been numbered incorrectly which will need addressing.

EDITOR COMMENTS

Reviewing Editor:

Thank you for your submission. Reviewers were enthusiastic about the potential influence of this paper, and noted important concerns that need to be addressed, as well as minor suggestions to improve clarity.

Senior Editor:

Thank you for submitting your manuscript for consideration by The Journal of Physiology. As part of the peer review process, we recruited two Referees with expertise in this field of study. Both Referees indicated that, if revised to address some major concerns, the manuscript has the potential to be impactful in this area of physiology. At this point, we would like to invite you to respond point-by-point to all Referee comments, revising the manuscript accordingly. While you are making revisions, please also provide details of the euthanasia approach used and bring the statistical reporting into compliance with The Journal policy. The Methods section currently has a very high degree of wording that is identical to published works, particularly a 2023 Nutrients paper. We would appreciate some attention to this section as well. Thank you again for your interest in The Journal of Physiology and we look forward to seeing your revised work.

If you choose to revise your manuscript, please make certain that it is in compliance with The Journal's statistics policy. For example, instead of SEM, variability needs to be represented as SD. There are also policies regarding reporting precise p-values (e.g., rather than $p < 0.05$) and representing individual data points on the graphs. Thank you.

REFEREE COMMENTS

Referee #1:

This manuscript by Chen et al. examines the impact of 10-week time-restricted feeding (TRF) on metabolic health in aged, female mice with obesity. Compared to the control group and high-fat fed ad libitum group, the high-fat fed TRF showed promising results, including attenuated weight gain and changes in fat deposition. Although the metabolic benefits of TRF have been studied previously, this work adds value by focusing on aged females. These findings may have important implications for translating TRF strategies to humans to reduce age-related risk of obesity and type 2 diabetes.

Although this paper is of interest, there are some limitations that if addressed would increase the enthusiasm for the paper. Suggestions to increase clarity (mostly minor) are below:

Major Limitations:

1. The rationale for using female mice requires further clarification. The manuscript references only one study using young male mice (Kohsaka et al., 2007). Consider expanding on why aged female models are particularly relevant and understudied. Here is a reference that may be helpful:

Chung, H., Chou, W., Sears, D. D., Patterson, R. E., Webster, N. J., & Ellies, L. G. (2016). Time-restricted feeding improves insulin resistance and hepatic steatosis in a mouse model of postmenopausal obesity. *Metabolism*, 65(12), 1743-1754.

Minor Suggestions:

1. Revise the figure legend numbering system. Currently, it jumps from Figure 3 back to Figure 2, and some figure titles repeat. Double check that figure numbers align with the text.
2. Address minor typos including: aing (page 4), Figure.1 B,C (page 10), impcts (page 17), and patters (page 17).
3. In the methods section, clarify that food was provided during the active (dark) period.

Referee #2:

This research paper examines the impact of time-restricted feeding (TRF) as a therapeutic approach to mitigate the negative effects of a high-fat diet in aged female mice. The novelty of this work lies in its focus on aged female mice and its inclusion of both metabolic measures and adipose tissue characteristics, such as adipocyte size and fat mass accumulation. This study addresses a gap in the literature, as little is known about depot-specific effects of TRF on adipose tissue metabolism in aged female mice with diet-induced obesity as most reported benefits have been described in male mice. Overall, this is a valuable contribution to the field; however, this reviewer offers several comments to improve the manuscript's clarity and readability.

General Comments:

1. Was this a high-fat only diet or was there also high-sugar? Could you please include the breakdown of the diet based on macros? This reviewer recommends a table or a description.
2. It would be helpful to clearly articulate how this study addresses diet-induced obesity in the context of older women or menopausal women.
3. It seemed like there were two experiments involving female mice, one with aged-female mice and one with middle-aged female mice (Figure 9). The discussion doesn't articulate the middle-aged female mice, and the methods do not describe them. This reviewer recommends an experimental timeline to explain both of these experiments.
4. It would help to include numbers on the side next to the lines to assist with readability for reviewers.
5. There are inconsistencies with numbers written as words and as numbers throughout the paper. Typically, numbers are written out as words when they are small or when they start a sentence while larger numbers are written as digits.

Abstract:

1. To improve readability and to highlight the important findings of this paper, this reviewer is suggesting a graphical abstract.

Introduction:

1. While time-restricted eating does limit the daily eating window, it is also known for being a temporal dietary intervention

that supports the circadian rhythm.

2. In the paragraph on page 4 that starts with "Adipose tissue is generally classified..." the next two sentences should have citations following them.
3. In the next two sentences beginning with "Under energy-deficient conditions..." this should also include citations.
4. In the sentence, "However, with prolonged HFD feeding...." There should be a space between the last word of the sentence and the citation.
5. The next paragraph has a typo where "Aing" is written instead of "Aging".
6. The last paragraph of the introduction describes why female mice were used, however, it would be beneficial to include some more data on females suffering from metabolic issues as they age and how TRF may ameliorate that. Is this study trying to understand underlying mechanisms of using TRF so that it can be applied to menopausal women that have higher incidences of obesity, heart disease, type 2 diabetes, etc.?

Methods:

1. This reviewer was wondering why there were no young female mice to serve as a baseline comparison between the middle-aged female mice and the older female mice.
2. A methodological timeline here would be very helpful.
3. Please include when lights were on and lights were off in the facility. It describes 12 hour light/dark cycles.
4. Please include when procedures were conducted using zeitgeber timing.
5. This paper describes using 18 14-18 month old C57BL/6J mice however in the next sentence 16 of them were fed HFD for 12 weeks but 6 of them were also used on an ad libitum normal chow diet. This doesn't make sense unless there were 22 older female mice. How many mice were placed into each of the groups following 12 weeks? How many mice were middle-aged and used for that experiment? How old were the middle-aged female mice? This needs more detail.
6. What time of day did the TRF group have access to food? Was it during the inactive or active cycle?
7. The mice were housed in groups of 3-4 per cage, how was individual food consumption calculated for each mouse?
8. Please include a description of the macro breakdown of both the high-fat diet and the normal chow. How much carbohydrates were in the diet? What type of high-fat was in the chow? Was it animal rennet or vegetable oils?
9. What time of day were the mice placed in the metabolic chambers and what time of day were glucose tolerance and insulin tolerance tests conducted.
10. How were the animals sacrificed?
11. Why were the final 2 days of food intake averaged to determine total food intake in grams/day? Why specifically 2 days?
12. Why was VCO₂ and RER measured over 3-days?
13. Was light amount measured for all these procedures? What were the Lux measures? Was this done during their active or inactive phase? Light at the wrong time of day can also cause circadian disruption.
14. Where was the 0.9% saline from? Was this made in house or ordered?
15. Please include the vendor for the Microtac Bluewave.
16. At the end of page 7, "Each sample was measured at least in duplicate." Does this mean that the samples were measured in duplicate or triplicate? Which samples were duplicated versus not?
17. Please describe how the bolus of glucose was made.
18. On page 8, there is a typo where it says, "The mice were fast for 4 hours." It should be the mice were fasted for 4 hours.
19. For the bolus of insulin, it states "using body weight (kg) x 0.75 insulin". What measurement was used? Was it L?
20. How was fasting blood glucose measured? Was it through tail vein? Please describe.
21. Please include more detail on the steps taken to dehydrate by ethanol solutions in the Hematoxylin and Eosin Staining

of Tissues.

22. Please define H&E and then use the abbreviation.

23. What was used to section the tissue?

24. What was the resinous mounting medium?

25. Which Leica microscope was used?

26. At the beginning of page 9, "Total RNA from frozen tissue" which tissue was used? Was this the adipose tissue? Was Total RNA extracted using only TRIZOL?

27. Please define Tbp before using the abbreviation.

28. Please include the catalog number for the enzymatic assay used for serum analysis.

29. Under Statistical Analysis, "Data was analyzed" should be "Data were analyzed..."

30. Instead of, "p-values less than 0.05 were considered to be significant." This reviewer recommends "Statistical significance was defined as a p-value less than 0.05." Or capitalize the P at the start of the sentence.

31. Can the authors please clarify why the data were analyzed by student t-test and one-way ANOVA?

32. Consider reporting mean +/- standard deviation (SD) for descriptive purposes and reserve SEM for inferential statistics. If the authors do wish to use standard error of the mean, please define it as standard error of the mean (SEM).

33. The time-series data appear to involve repeated measures; a repeated measures ANOVA or mixed-effects model might be more appropriate than a one-way ANOVA.

34. What post-hoc tests were used to determine which groups differ?

Results:

1. Please include the amount of animals in each group used in all figure captions for each group.

2. On Page 17, there is a typo at the start of this sentence "To determine whether aging impcts..." it should be "impacts".

3. The middle-aged female mice experiment was not described in the methods.

Discussion:

1. This reviewer would be interested to know if there are differences in hormone levels between the middle-aged and older female mice. Were sex hormones measured in these two groups to explain some of the effects seen?

2. Were there any limitations to this study?

3. How old are middle-aged mice in the previous studies cited (Hepler et al., 2022) Were they the same age as the middle-aged mice used in this study?

4. While the paper discussed age-related decline of long-term HFD exposure, this reviewer recommends discussing how this translates to humans, specifically, using older female aged mice and how that may be translated to menopausal women.

Conclusion:

1. The last sentence, "for improving metabolic health in old female mice..." should be "older female mice".

2. TRF is a promising dietary approach in older female mice or in HFD-fed older female mice?

3. It would be beneficial to connect this study to how western diets are typically higher in fat content and how this has led to an increase in obesity and metabolic disorders. It would help to tie in more of the introduction describing why female aged mice were studied with the Western diet and how this study could add to the understanding of underlying mechanisms to translate this to human studies.

END OF COMMENTS

UNIVERSITY OF MINNESOTA

Twin Cities Campus

Department of Food Science and Nutrition
*College of Food, Agricultural and
Natural Resource Sciences*

*Food Science and Nutrition
1334 Eckles Avenue Rm 225
St. Paul, MN 55108-1038 USA
612-624-1290
Fax: 612-625-5272
<http://fscn.cfans.umn.edu>*

September 26, 2025

Dear Editor:

We sincerely appreciate the opportunity to revise our manuscript (**JP-RP-2025-289464**) and would like to thank the reviewers for their constructive and insightful comments.

In response to the reviewers' feedback, we made substantial revisions to the abstract, introduction, results and discussion sections. Additionally, we have carefully addressed editor's comments and each reviewer's comments point by point, as detailed below, and have highlighted all revisions using Track Changes in the manuscript.

We believe that these revisions have significantly improved the manuscript, and we hope it will now be suitable for publication.

Sincerely,

Xiaoli Chen, M.D., Ph.D.
Professor and General Mills Chair
in Genomics for Healthful Foods
Department of Food Science and Nutrition
University of Minnesota

JP-RP-2025-289464R1 The Journal of Physiology Returned to Authors with Queries on Oct. 1, 2025

Before we process the paper, we ask that you provide the following required items:

1) - Please ensure that any tables are editable and in Word format.

Response: Table 1 has been replaced with an editable version.

JP-RP-2025-289464R1 The Journal of Physiology Returned to Authors with Queries on Sept. 30, 2025

Before we process the paper, we ask that you provide the following required items:

1) - Ethical approval. Please state the agent, route and dose of anaesthetic prior to euthanasia, or include a statement that no anaesthetic was used and why.

Response: This has been revised as follows:

“Animal euthanasia procedure: anesthetics were not used for animal euthanasia in this study. Instead, carbon dioxide (CO₂) inhalation was employed, in accordance with protocols approved by the IACUC at the University of Minnesota. Compressed CO₂ gas was supplied from cylinders and regulated using a pressure-reducing regulator and flow meter. Animals were placed in a euthanasia chamber, and 100% CO₂ was introduced at a fill rate of 30-70% of the chamber volume per minute to displace ambient air. Unconsciousness was confirmed by the absence of respiration and the presence of corneal opacification. CO₂ flow was maintained for at least one minute beyond the cessation of respiration to ensure complete euthanasia. Following this, cardiac puncture was performed for blood collection. When necessary, cervical dislocation was used as a secondary method to confirm death and facilitate tissue collection.”

2) - Please ensure that all figures and tables have a title and legend, and that they have been cited within the main article text. Table 1 is currently missing a legend.

Response: the legend for Table 1 has been added as follows.

“Forward (F) and reverse (R) primer sequences for each gene are listed along with gene name and gene symbol. Primers were validated for specificity by melt curve analysis.”

JP-RP-2025-289464R1 The Journal of Physiology Returned to Authors with Queries on Sept 29, 2025

Before we process the paper, we ask that you provide the following required items:

1) Please provide details of the anaesthesia used - agent, dose and route of administration.

For more information see The Journal's policies regarding animal experiments

(<https://physoc.onlinelibrary.wiley.com/hub/animal-experiments>) and adherence to these policies must be stated in the manuscript.

Response: We have added the following information the Methodes – Ethical Approval section:

“Animals were euthanized at the end of the study using CO₂ inhalation. Compressed CO₂ gas was supplied from cylinders and regulated by a pressure-reducing regulator and flow meter. Animals were placed in the chamber, and 100% CO₂ was introduced at a fill rate of 30-70% displacement of the chamber volume per minute, allowing CO₂ to displace the existing air. Loss of unconsciousness was confirmed by the absence of respiration and the presence of corneal

opacification. CO₂ flow was then maintained for at least one minute after respiration ceased. Following euthanasia, cardiac puncture was performed for blood collection, and cervical dislocation was carried out as a secondary method of euthanasia when necessary.”

2) - Your paper contains Supporting Information of a type that we no longer publish, including supplementary tables and figures. Any information essential to an understanding of the paper must be included as part of the main manuscript and figures. The only Supporting Information that we publish are video and audio, 3D structures, program codes and large data files. Your revised paper will be returned to you if it does not adhere to our Supporting Information Guidelines.

Response: The table S1 has been removed from Supplementary materials to the main article as Table 1

3) - Please include an Abstract Figure Legend within the main article file.

Response: An Abstract Figure Legend has been included within the main article file

“Abstract Figure. A 10-week TRF regimen was implemented in aged female mice following 12 weeks of HFD feeding. TRF partially reversed HFD-induced weight and fat mass gain, reduced adipocyte size, and increased size heterogeneity in white adipose tissue. It also enhanced energy expenditure, lowered RER (particularly during the light phase), decreased fasting blood glucose, and reduced hepatic lipid accumulation. At the molecular level, TRF promoted metabolic remodeling in adipose tissue, including upregulation of genes related to adipogenesis and lipid cycling, with depot-specific changes in mitochondrial oxidation and circadian rhythm gene expression.”

4) - Please ensure that all figures and tables have a title and legend, and that they have been cited within the main article text. Figure 1 is currently missing a legend.

Response: Figure 1 legend has been added within the main article file.

“Figure 1. Experimental timeline. *In the aged cohort*, 22 aged female mice (14-18-month-old) mice were used. Six mice remained on an ad libitum normal chow diet (Control), while 16 were fed a high-fat diet (HFD) for 12 weeks to induce obesity. The HFD-fed mice were then divided into two groups: ad libitum HFD (HFD-AL, n=8) or time-restricted feeding (HFD-TRF, n=8). In a parallel study, 11 middle aged female mice (3 months old) were assigned to chow (n=4), HFD-AL (n=4), or HFD-TRF (n=3). After 12 weeks of HFD, they reached 6 months of age and continued on HFD-AL or HFD-TRF for an additional 10 weeks. Mice were housed 3–4 per cage under 12 h light/dark cycles (lights off 8PM-8AM), with water ad libitum. TRF groups had food access for 10h during the dark phase (8:30PM-6:30AM). At weeks 6-7 of TRF, GTT and ITT were performed; indirect calorimetry and meal pattern analysis were conducted at week 8. At week 10, mice were euthanized for blood and tissue collection.”

REVISION CHECKLIST:

REQUIRED ITEMS

1) - Include a Key Points list in the article itself, before the Abstract.

Response: This information has been added to the manuscript.

2) - Author photo and profile. First or joint first authors are asked to provide a short biography (no more than 100 words for one author or 150 words in total for joint first authors) and a portrait photograph. These should be uploaded and clearly labelled together in a Word document with the revised version of the manuscript. See Information for Authors for further details.

Response: the author's photo and profile has been provided.

3) - You must start the Methods section with a paragraph headed Ethical approval (https://jp.msubmit.net/cgi-bin/main.plex?form_type=display_requirements#methods). Research must comply with The Journal's policies regarding animal experiments (<https://physoc.onlinelibrary.wiley.com/hub/animal-experiments>) and adherence to these policies must be stated in the manuscript.

Authors should confirm in their Methods section that their experiments were carried out according to the guidelines laid down by their institution's animal welfare committee, including an ethics approval reference number. The Methods section must contain a statement about access to food, water and housing, details of the anaesthetic regime: anaesthetic used, dose and route of administration, and method of killing the experimental animals.

Response: The relevant information has been provided.

4) - Your manuscript must include a complete Additional Information section, including competing interests; funding; author contributions and acknowledgements.

Response: The information regarding competing interests; funding; author contributions and acknowledgements has been added.

5) - Please upload separate high-quality figure files via the submission form.

6) - Please ensure that any tables are editable and in Word format, and wherever possible,

embedded in the article file itself.

7) - Please ensure that the Article File you upload is a Word file.

8) - Papers must comply with the Statistics Policy: https://jp.msubmit.net/cgi-bin/main.plex?form_type=display_requirements#statistics.

Response: The revision has been made to comply with the Statistics Policy.

9) - A Data Availability Statement is required for all papers reporting original data. This must be in the Additional Information section of the manuscript itself. It must have the paragraph heading 'Data Availability Statement'. All data supporting the results in the paper must be either: in the paper itself; uploaded as Supporting Information for Online Publication; or archived in an appropriate public repository. The statement needs to describe the availability or the absence of shared data. Authors must include in their statement: a link to the repository they have used, or a statement that it is available as Supporting Information; reference the data in the appropriate section(s) of their manuscript; and cite the data they have shared in the References section. Whenever possible, the scripts and other artefacts used to generate the analyses presented in the paper should also be publicly archived. If sharing data compromises ethical standards or legal requirements then authors are not expected to share it, but must note this in their statement. For more information, see our Statistics Policy.

Response: This has been updated.

10) - Please include an Abstract Figure file, as well as the Figure Legend text within the main article file. The Abstract Figure is a piece of artwork designed to give readers an immediate understanding of the research and should summarise the main conclusions. If possible, the image should be easily 'readable' from left to right or top to bottom. It should show the physiological relevance of the manuscript so readers can assess the importance and content of its findings. Abstract Figures should not merely recapitulate other figures in the manuscript. Please try to keep the diagram as simple as possible and without superfluous information that may distract from the main conclusion(s). Abstract Figures must be provided by authors no later than the revised manuscript stage and should be uploaded as a separate file during online submission labelled as File Type 'Abstract Figure'. Please also ensure that you include the figure legend in the main article file. All Abstract Figures should be created using BioRender. Authors should use The Journal's premium BioRender account to export high-resolution images. Details on how to use and access the premium account are included as part of this email.

Response: An Abstract Figure file has been included.

11)- Please ensure that all figures and tables have a title and legend, and that they have been cited within the main article text. The Figure legends have been numbered incorrectly

which will need addressing.

Response: This has been fixed.

EDITOR COMMENTS

Reviewing Editor:

Thank you for your submission. Reviewers were enthusiastic about the potential influence of this paper, and noted important concerns that need to be addressed, as well as minor suggestions to improve clarity.

Senior Editor:

Thank you for submitting your manuscript for consideration by The Journal of Physiology. As part of the peer review process, we recruited two Referees with expertise in this field of study. Both Referees indicated that, if revised to address some major concerns, the manuscript has the potential to be impactful in this area of physiology. At this point, we would like to invite you to respond point-by-point to all Referee comments, revising the manuscript accordingly. While you are making revisions, please also provide details of the euthanasia approach used and bring the statistical reporting into compliance with The Journal policy. The Methods section currently has a very high degree of wording that is identical to published works, particularly a 2023 Nutrients paper. We would appreciate some attention to this section as well. Thank you again for your interest in The Journal of Physiology and we look forward to seeing your revised work.

If you choose to revise your manuscript, please make certain that it is in compliance with The Journal's statistics policy. For example, instead of SEM, variability needs to be represented as SD. There are also policies regarding reporting precise p-values (e.g., rather than $p < 0.05$) and representing individual data points on the graphs. Thank you.

Response: The methods section has been substantially revised. In accordance with the Journal's statistics policy, SEM has been replaced with SD, and exact p-values have been incorporated into either the figures or their corresponding legends for all data presentations.

REFEREE COMMENTS

Referee #1:

Major Limitations:

1. The rationale for using female mice requires further clarification. The manuscript references only one study using young male mice (Kohsaka et al., 2007). Consider expanding on why aged female models are particularly relevant and understudied. Here is a reference that may be helpful:

Chung, H., Chou, W., Sears, D. D., Patterson, R. E., Webster, N. J., & Ellies, L. G. (2016). Time-restricted feeding improves insulin resistance and hepatic steatosis in a mouse model of postmenopausal obesity. *Metabolism*, 65(12), 1743-1754.

Response: The recommended paper has been cited and the Introduction section has been updated as follows: “Although multiple studies have reported the metabolic benefits of TRF in young male mice with evidence suggesting each adipose depot responds differently to a TRF regimen (Bushman et al., 2023), there is limited understanding of its depot-specific effects on adipose tissue metabolism in aged female mice with diet-induced obesity. Importantly, one study using a postmenopausal female mouse model fed a HFD demonstrated that TRF led to rapid weight loss, improved glucose tolerance, and reduced hepatic lipid accumulation, accompanied by a shift in hepatic metabolism from gluconeogenesis toward ketogenesis (Chung et al., 2016). This is particularly relevant because postmenopausal women experience unique metabolic risks, including increased visceral adiposity, insulin resistance, and susceptibility to nonalcoholic fatty liver disease (Chung et al., 2015). Yet, most preclinical metabolic studies have relied on young male rodents, overlooking sex- and age-related differences in disease pathophysiology and therapeutic response. This gap in research limits the translational relevance of findings for older women, who represent a high-risk and growing patient population.”

Minor Suggestions:

1. Revise the figure legend numbering system. Currently, it jumps from Figure 3 back to Figure 2, and some figure titles repeat. Double check that figure numbers align with the text.

Response: I think this has been fixed. On my version it is showing it to be the correct order and the titles differ by adipose tissue depot as well.

2. Address minor typos including: aing (page 4), Figure.1 B,C (page 10), impcts (page 17), and patters (page 17).

Response: These have been corrected, thank you.

3. In the methods section, clarify that food was provided during the active (dark) period.

Response: This has been changed to the following, thank you. “ The AL groups had constant access to food and the TRF group had access to food 10 h/day during the active (dark) period.”

Referee #2:

General Comments:

1. Was this a high-fat only diet or was there also high-sugar? Could you please include the breakdown of the diet based on macros? This reviewer recommends a table or a description.

Response: This diet contains a higher percentage of fat and reduced percentage of carbohydrates. We have included the following description of its nutritional profile from the manufacture in the Methods section of the manuscript. “According to the manufacture (<https://www.bio-serv.com/product/HFPellets.html>), the HFD contains 20.5% protein, 36% fat, and 35.7% carbohydrate compared to the normal diet which contains 20.5% protein, 7.2% fat, and 61.6% carbohydrate. Regarding the type of fat, the diet contains lard as its primary fat source.”

2. It would be helpful to clearly articulate how this study addresses diet-induced obesity in the context of older women or menopausal women.

Response: The introduction has been updated as follows. “Although multiple studies have reported the metabolic benefits of TRF in young male mice with evidence suggesting each adipose depot responds differently to a TRF regimen (Bushman et al., 2023), there is limited understanding of its depot-specific effects on adipose tissue metabolism in aged female mice with diet-induced obesity. Importantly, one study using a postmenopausal female mouse model fed a HFD demonstrated that TRF led to rapid weight loss, improved glucose tolerance, and reduced hepatic lipid accumulation, accompanied by a shift in hepatic metabolism from gluconeogenesis toward ketogenesis (Chung et al., 2016). This is particularly relevant because postmenopausal women experience unique metabolic risks, including increased visceral adiposity, insulin resistance, and susceptibility to nonalcoholic fatty liver disease (Chung et al., 2015). Yet, most preclinical metabolic studies have relied on young male rodents, overlooking sex- and age-related differences in disease pathophysiology and therapeutic response. This gap in research limits the translational relevance of finding for older women, who represent a high-risk and growing patient population.”

3. It seemed like there were two experiments involving female mice, one with aged-female mice and one with middle-aged female mice (Figure 9). The discussion doesn't articulate the middle-aged female mice, and the methods do not describe them. This reviewer recommends an experimental timeline to explain both of these experiments.

Response: Thank you. This information has been added to the methods section, along with the number of mice in each group and a figure (Figure 1) showing an experimental timeline for both studies.

4. It would help to include numbers on the side next to the lines to assist with readability for reviewers.

Response: This has been added, thank you.

5. There are inconsistencies with numbers written as words and as numbers throughout the paper. Typically, numbers are written out as words when they are small or when they start a sentence while larger numbers are written as digits.

Response: This has been updated within the Methods section. Thank you.

Abstract:

1. To improve readability and to highlight the important findings of this paper, this reviewer is suggesting a graphical abstract.

Response: We appreciate the suggestion and have included a graphical abstract (abstract figure).

Introduction:

1. While time-restricted eating does limit the daily eating window, it is also known for being a temporal dietary intervention that supports the circadian rhythm.

Response: We appreciate the addition, this is been mentioned.

2. In the paragraph on page 4 that starts with "Adipose tissue is generally classified..." the next two sentences should have citations following them.

Response: Thank you, this has been edited, and three more references have been cited.

References:

Cedikova, M., Kripnerová, M., Dvorakova, J., Pitule, P., Grundmanova, M., Babuska, V., Mullerova, D., and Kuncova, J. (2016). Mitochondria in White, Brown, and Beige Adipocytes. *Stem Cells Int.* 2016.

Giralt, M., and Villarroya, F. (2013). White, brown, beige/brite: different adipose cells for different functions? *Endocrinology* 154, 2992–3000.

Peirce, V., Carobbio, S., and Vidal-Puig, A. (2014). The different shades of fat. *Nature* 510, 76–83.

3. In the next two sentences beginning with "Under energy-deficient conditions..." this should also include citations.

Response: Thank you for the suggestion. The statement regarding lipolysis under energy-deficient conditions and adipose tissue expansion describes well-established physiological processes and is commonly presented without citation. Therefore, we have not added a specific reference. However, we have cited a reference for the description of "Excess caloric consumption leads to adipose tissue expansion via both hyperplasia and hypertrophy mechanisms"

Ref: White, U. (2023). Adipose tissue expansion in obesity, health, and disease. *Front. Cell Dev. Biol.* 11, 1188844.

4. In the sentence, "However, with prolonged HFD feeding..." There should be a space between the last word of the sentence and the citation.

Response: This has been corrected, and a reference has been cited. Thank you.

5. The next paragraph has a typo where "Aing" is written instead of "Aging".

Response: This has been corrected, thank you.

6. The last paragraph of the introduction describes why female mice were used, however, it would be beneficial to include some more data on females suffering from metabolic issues as they age and how TRF may ameliorate that. Is this study trying to understand underlying mechanisms of using TRF so that it can be applied to menopausal women that have higher incidences of obesity, heart disease, type 2 diabetes, etc.?

Response: Thanks for the reviewer's suggestion! We have updated this paragraph in the introduction section as follows. "Although multiple studies have reported the metabolic benefits of TRF in young male mice with evidence suggesting each adipose depot responds differently to a TRF regimen (Bushman et al., 2023), there is limited understanding of its depot-specific effects on adipose tissue metabolism in aged female mice with diet-induced obesity. Importantly, one study using a postmenopausal female mouse model fed a HFD demonstrated that TRF led to rapid weight loss, improved glucose tolerance, and reduced hepatic lipid accumulation, accompanied by a shift in hepatic metabolism from gluconeogenesis toward ketogenesis (Chung et al., 2016). This is particularly relevant because postmenopausal women experience unique metabolic risks, including increased visceral adiposity, insulin resistance, and susceptibility to nonalcoholic fatty liver disease (Chung et al., 2015). Yet, most preclinical metabolic studies have relied on young male rodents, overlooking sex- and age-related differences in disease pathophysiology and

therapeutic response. This gap in research limits the translational relevance of finding for older women, who represent a high-risk and growing patient population.”

Methods:

1. This reviewer was wondering why there were no young female mice to serve as a baseline comparison between the middle-aged female mice and the older female mice.

Response: Typically, mice need to be 8-12 weeks old to ensure metabolic maturity prior to starting a HFD. In our study, the middle-aged mice began the HFD at 12 weeks of age, which is considered the onset of middle age. After completing 12 weeks on the HFD, these mice reached middle age.

2. A methodological timeline here would be very helpful.

Response: We have added a diagram as Figure 1, showing a methodological timeline.

3. Please include when lights were on and lights were off in the facility. It describes 12 hour light/dark cycles.

Response: This has been added to the paper. “The mice were housed in groups of 3–4 per cage, with water ad libitum and in 12 h light/dark cycles with lights off from 12-24 ZT (8PM-8AM).”

4. Please include when procedures were conducted using zeitgeber timing.

Response: This has been added to the paper. “The mice were housed in groups of 3–4 per cage, with water ad libitum and in 12 h light/dark cycles with lights off from 12-24 ZT (8PM-8AM).”

5. This paper describes using 18 14-18 month old C57BL/6J mice however in the next sentence 16 of them were fed HFD for 12 weeks but 6 of them were also used on an ad libitum normal chow diet. This doesn't make sense unless there were 22 older female mice. How many mice were placed into each of the groups following 12 weeks? How many mice were middle-aged and used for that experiment? How old were the middle-aged female mice? This needs more detail.

Response: This has been updated in the paper as follows. “The study included 22 aged (14-18-month-old) female C57BL/6 mice (The Jackson Laboratory, Bar Harbor, ME, USA). Sixteen mice were fed a high-fat diet (HFD) for 12 weeks to induce obesity while 6 mice remained on an ad libitum normal chow diet (Control). After 12 weeks, HFD-fed mice were divided into two groups: 8 continued on ad libitum HFD (HFD-AL) group and 8 were switched to time-restricted feeding (HFD-TRF) group. In addition, 11 female mice at 3 months of age were enrolled in a middle-aged study. These consisted of four chow-fed

controls, four HFD-AL, and three HFD-TRF. After 12 weeks of HFD feeding, these mice reached 6 months of age and were either maintained on ad libitum HFD (HFD-AL) or switched to time-restricted feeding (HFD-TRF) for additional 10 weeks.”

6. What time of day did the TRF group have access to food? Was it during the inactive or active cycle?

Response: “The HFD-TRF group had food available from 8:30PM-6:30AM” has been added, during the active cycle.

7. The mice were housed in groups of 3-4 per cage, how was individual food consumption calculated for each mouse?

Response: As mentioned in the *Indirect Calorimetry and Body Weight* section on page 6, mice were individually housed in metabolic cages for the assessments of indirect calorimetry and food intake. Food intake was measured over a 5-day period during this time period.

8. Please include a description of the macro breakdown of both the high-fat diet and the normal chow. How much carbohydrates were in the diet? What type of high-fat was in the chow? Was it animal rennet or vegetable oils?

Response: As noted in our response to the general comments -question #1, this diet contains a higher percentage of fat and reduced percentage of carbohydrates. We have included the manufacturer’s description of its nutritional profile from the manufacture in the manuscript. “According to the manufacture (<https://www.bio-serv.com/product/HFPellets.html>), the HFD contains 20.5% protein, 36% fat, and 35.7% carbohydrate compared to the normal diet which contains 20.5% protein, 7.2% fat, and 61.6% carbohydrate. Regarding the type of fat, the diet contains lard as its primary fat source.”

9. What time of day were the mice placed in the metabolic chambers and what time of day were glucose tolerance and insulin tolerance tests conducted.

Response: They were placed in the metabolic chamber at 8am for 3 days. GTT and ITT tests were conducted between 10am-12pm on separate weeks.

10. How were the animals sacrificed?

Response: This has been added into the methods section. “Animals were euthanized at the end of the study via CO2 inhalation. Once animals were unresponsive, cardiac puncture was performed as a secondary method to ensure death and for blood collection.”

11. Why were the final 2 days of food intake averaged to determine total food intake in grams/day? Why specifically 2 days?

Response: Mice were single-housed for 5 days to allow acclimation to the cage and food measurement apparatus; daily food intake was recorded each day and grams/day intake was calculated as the average of the final 2 days (days 4–5) to capture stabilized feeding after their initial adjustment (day 1-3).

12. Why was VCO₂ and RER measured over 3-days?

Response: This is how our Department of Integrative Biology and Physiology at the University of Minnesota does this portion with the indirect calorimetry for the similar reason described in the Response to the question #11 above.

13. Was light amount measured for all these procedures? What were the Lux measures? Was this done during their active or inactive phase? Light at the wrong time of day can also cause circadian disruption.

Response: The light amount was not measured during these procedures.

14. Where was the 0.9% saline from? Was this made in house or ordered?

Response: This was made in house.

15. Please include the vendor for the Microtac Bluewave.

Response: Optical Particle Analyze (Model: Microtrac SIA) has been added to the manuscript

16. At the end of page 7, "Each sample was measured at least in duplicate." Does this mean that the samples were measured in duplicate or triplicate? Which samples were duplicated versus not?

Response: We did duplicates for each sample. The duplicate measurements were included to help account for variance within a given sample, as some samples contained slightly higher particle concentrations than others. The suspended osmium particles varied in total amounts, so duplicate readings ensured greater reliability of the reported values.

17. Please describe how the bolus of glucose was made.

Response: Thank you. This has been added to the methods. "Bolus of glucose (0.75 g/kg body weight) was prepared from a filter-sterilized D-glucose solution (300 mg/mL in saline) and administered intraperitoneally at 2.5 µl per gram body weight."

18. On page 8, there is a typo where it says, "The mice were fast for 4 hours." It should be the mice were fasted for 4 hours.

Response: This has been corrected, thank you.

19. For the bolus of insulin, it states "using body weight (kg) x 0.75 insulin". What measurement was used? Was it μL ?

Response: Thank you. This has been added to the methods. "A bolus of insulin (0.75 U/kg body weight) was prepared from a diluted stock solution of regular human insulin (Humulin R; Lilly NDC 0002-8215-01)."

20. How was fasting blood glucose measured? Was it through tail vein? Please describe.

Response: Fasting BG was measured from the tail vein. The initial drop of blood was discarded before collecting the subsequent drop for analysis with a handheld glucometer.

21. Please include more detail on the steps taken to dehydrate by ethanol solutions in the Hematoxylin and Eosin Staining of Tissues.

Response: We modified the description of the H&E method as follows. "Tissues were fixed in 10% neutral buffered formalin (VWR International, LLC, Radnor, PA, USA) for 2-3 days, then dehydrated by 70% ethanol solutions for 3-5 days and processed for embedding in paraffin. Tissue samples were Hematoxylin and Eosin (H&E) stained using a standard protocol at the University of Minnesota Histology Core."

Since the H&E method is well established and conducted by the core facility and the protocol is standardized, we don't think the detailed steps are necessary.

22. Please define H&E and then use the abbreviation.

Response: This has been corrected, thank you.

23. What was used to section the tissue?

Response: We sent our samples to the University of Minnesota Histology Core, where the core staff performed the H&E staining. Since this method is well established, we assume that the protocol used by the core is standardized. Unfortunately, we don't have specific detailed information regarding the tissue sectioning and processing procedure.

24. What was the resinous mounting medium?

Response: We sent our samples to the University of Minnesota Histology Core, where the core staff performed the H&E staining. Since this method is well established, we assume that the protocol used by the core is standardized. Unfortunately, we don't have specific detailed information regarding the tissue sectioning and processing procedure.

25. Which Leica microscope was used?

Response: it is a Leica DM IL microscope. This information has been added to the paper.

26. At the beginning of page 9, "Total RNA from frozen tissue" which tissue was used? Was this the adipose tissue? Was Total RNA extracted using only TRIZOL?

Response: It was stated on lines 214-217 which tissues. "Tissue section from brown adipose tissue and liver were used for histological analysis, while small portions of inguinal and epididymal adipose tissue were used for both histological analysis and fat cell sizing and the remaining tissue was snap-frozen via liquid nitrogen and stored for later analysis." Additionally, yes, TRIZOL was used, "Total RNA was extracted from frozen tissue using TRIZOL reagent (Invitro, Carlsbad, CA, USA) and treated with DNAase to remove genomic DNA prior to cDNA synthesis using Superscript II reverse transcription kit (Invitrogen, Carlsbad, CA, USA)."

27. Please define Tbp before using the abbreviation.

Response: This has been updated. Thank you. "For quantification TATA-box binding protein (Tbp) mRNA served as an endogenous control within inguinal and brown adipose tissue. The primer sequences for amplifying the target genes are shown in Table 1."

28. Please include the catalog number for the enzymatic assay used for serum analysis.

Response: This has been updated. Thank you. "Serum triglyceride level was determined using Triglycerides Enzymatic Assay Kit (Stanbio Laboratory, Boerne, TX, USA; #2100430). Serum free fatty acids and β -hydroxybutyrate levels were determined using Free Fatty Acid Quantitation Kit (Sigma-Aldrich, # MAK044) and β -Hydroxybutyrate Assay Kit (Sigma-Aldrich, # MAK041) according to the manufacturers' instructions."

29. Under Statistical Analysis, "Data was analyzed" should be "Data were analyzed...".

Response: This has been updated, thank you.

30. Instead of, "p-values less than 0.05 were considered to be significant." This reviewer recommends "Statistical significance was defined as a p-value less than 0.05." Or capitalize the P at the start of the sentence.

Response: This has been updated, thank you.

31. Can the authors please clarify why the data were analyzed by student t-test and one-way ANOVA?

Response: Results were expressed as mean \pm SEM. Data from aged and middle-aged female mice were analyzed using Student's *t*-test when comparing two groups, and one-way ANOVA when comparing more than two groups, in GraphPad Prism (version Prism 9.4.1).

32. Consider reporting mean \pm standard deviation (SD) for descriptive purposes and reserve SEM for inferential statistics. If the authors do wish to use standard error of the mean, please define it as standard error of the mean (SEM).

Response: This has been changed to standard deviation (SD) instead of SEM for all the figures. Thank you.

33. The time-series data appear to involve repeated measures; a repeated measures ANOVA or mixed-effects model might be more appropriate than a one-way ANOVA.

Response: Thank you for the suggestion. However, our primary interest lies comparing groups at individual time points rather than analyzing trajectories over time, as the animal's responses to diet or TRF differ between light and dark cycles. Our goal was to examine group differences at each specific time point, particularly across the light and dark cycles, rather than to model within-subject changes over time. For this reason, we used one-way ANOVA at each time point, which directly aligns with and addresses our research question.

34. What post-hoc tests were used to determine which groups differ?

Response: Student *t* test was used as post hoc analysis to determine which groups differ.

Results:

1. Please include the amount of animals in each group used in all figure captions for each group.

Response: This information has been added to the figure legends

2. On Page 17, there is a typo at the start of this sentence "To determine whether aging impacts..." it should be "impacts".

Response: Updated.

3. The middle-aged female mice experiment was not described in the methods.

Response: This has been updated with numbers in each group for the middle-aged female mice. The methods were the exact same, minus GTT/ITT and metabolic cages.

Discussion:

1. This reviewer would be interested to know if there are differences in hormone levels between the middle-aged and older female mice. Were sex hormones measured in these two groups to explain some of the effects seen?

Response: It would be interesting. However, we did not measure sex hormone levels in this manuscript.

2. Were there any limitations to this study?

Response: The following limitations were added to the conclusion section in the manuscript. "However, one limitation of this study is the lack of comparison between the beneficial effects of TRF in young versus aged female mice. Additionally, we did not examine the impact of TRF in post-menopausal females under normal dietary conditions."

3. How old are middle-aged mice in the previous studies cited (Hepler et al., 2022) Were they the same age as the middle-aged mice used in this study?

Response: They were three months old, which is similar to our mice used. This has been included in the manuscript.

4. While the paper discussed age-related decline of long-term HFD exposure, this reviewer recommends discussing how this translates to humans, specifically, using older female aged mice and how that may be translated to menopausal women.

Response: This is a good point! We have answered similar questions and added the relevant information to the introduction section.

Conclusion:

1. The last sentence, "for improving metabolic health in old female mice..." should be "older female mice".

Response: Updated.

2. TRF is a promising dietary approach in older female mice or in HFD-fed older female mice?

Response: Meant to say HFD-fed older female mice. Updated.

3. It would be beneficial to connect this study to how western diets are typically higher in fat content and how this has led to an increase in obesity and metabolic disorders. It would help to tie in more of the introduction describing why female aged mice were studied with

the Western diet and how this study could add to the understanding of underlying mechanisms to translate this to human studies.

Response: One additional sentence has been added to the conclusion to tie this back into western diets. “Given that Western diets are typically higher in fat and strongly associated with rising rates of obesity and metabolic disorders, this model provides important insight into how altered feeding patterns can counteract diet-induced metabolic dysfunction.”

Dear Dr Chen,

Re: JP-RP-2025-289464R1 "**Impact of time-restricted feeding on metabolic health and adipose tissue metabolism in aged female mice with high fat diet-induced obesity**" by Xiaoli Chen, Theresa Bushman, and Hongming Su

Thank you for submitting your manuscript to The Journal of Physiology. It has been assessed by a Reviewing Editor and by 2 expert referees and we are pleased to tell you that it is acceptable for publication following satisfactory revision.

REVISION CHECKLIST:

We look forward to receiving your revised submission.

Yours sincerely,

Karyn Hamilton
Senior Editor
The Journal of Physiology

REQUIRED ITEMS

1) - Papers must comply with the Statistics Policy: https://jp.msubmit.net/cgi-bin/main.plex?form_type=display_requirements#statistics.

In summary:

- If n {less than or equal to} 30, all data points must be plotted in the figure in a way that reveals their range and distribution. A bar graph with data points overlaid, a box and whisker plot or a violin plot (preferably with data points included) are acceptable formats.
- If $n > 30$, then the entire raw dataset must be made available either as supporting information, or hosted on a not-for-profit repository, e.g. FigShare, with access details provided in the manuscript.
- 'n' clearly defined (e.g. x cells from y slices in z animals) in the Methods. Authors should be mindful of pseudoreplication.
- All relevant 'n' values must be clearly stated in the main text, figures and tables.
- The most appropriate summary statistic (e.g. mean or median and standard deviation) must be used. Standard Error of the Mean (SEM) alone is not permitted.
- Exact p values must be stated. Authors must not use 'greater than' or 'less than'. Exact p values must be stated to three significant figures even when 'no statistical significance' is claimed.

EDITOR COMMENTS

Senior Editor:

Thank you for your manuscript revisions. The Referees were pleased with the result of your efforts and at this time we are happy to Provisionally Accept the manuscript, pending satisfactory final revisions. We appreciated the work you did to bring the manuscript into compliance with policies of The Journal. In particular, we thank you for the revisions you made to bring the manuscript closer to compliance with The Journal's Statistics Policy. It seems that there is still some revision to be done. The policy requires representation of individual data points on the graphs when possible. Please revisit this statistics policy, making revisions to the manuscript. We look forward to seeing the final revisions and thank you for your interest in The Journal of Physiology!

Thank you for the revisions you made to bring the manuscript closer to compliance with The Journal's Statistics Policy. It seems that there is still some revision to be done. The policy requires representation of individual data points on the graphs when possible.

Reviewing Editor:

Thank you very much for this responsive resubmission. Both reviewers were enthusiastic and noted that the manuscript is

much improved and likely to be highly influential in the field.

REFEREE COMMENTS

Referee #1:

Thank you for the revisions and responsiveness to the feedback. This manuscript has improved by addressing the importance of studying aged female rodents. This addresses a major gap in the previous literature. The current version of this manuscript now notes that previous studies have focused on time-restricted feeding in young male mice. The methods have also been clarified. Details about the food type and availability as well as the animal euthanasia procedures help clarify the experimental design and improve reproducibility. Overall, this manuscript provides significant insights into the physiological mechanisms of time-restricted eating in a understudied, high-risk population.

Referee #2:

Thank you for carefully addressing the previous comments and for the thoughtful revisions. The changes have clearly improved the clarity and scientific quality of the manuscript.

END OF COMMENTS

UNIVERSITY OF MINNESOTA

Twin Cities Campus

*Department of Food Science and Nutrition
College of Food, Agricultural and
Natural Resource Sciences*

*Food Science and Nutrition
1334 Eckles Avenue Rm 225
St. Paul, MN 55108-1038 USA
612-624-1290
Fax: 612-625-5272
<http://fscn.cfans.umn.edu>*

December 10, 2025

Dear Editor:

We sincerely appreciate the opportunity to revise our manuscript (**JP-RP-2025-289464R1**) and thank the reviewers for their positive feedback.

In response to the editor's comments, we have incorporated the suggested changes to comply with the Statistics Policy. Also, we have provided a point-by-point response to editor's comments below and highlighted all revisions in the manuscript using Track Changes.

We hope this revised version of the manuscript meets the requirements and is now suitable for publication.

Sincerely,

Xiaoli Chen, M.D., Ph.D.
Professor and General Mills Chair
in Genomics for Healthful Foods
Department of Food Science and Nutrition
University of Minnesota

RESPONSE TO EDITOR'S COMMENTS

REQUIRED ITEMS

1) - Papers must comply with the Statistics Policy: https://jp.msubmit.net/cgi-bin/main.plex?form_type=display_requirements#statistics.

In summary:

- If $n \leq 30$, all data points must be plotted in the figure in a way that reveals their range and distribution. A bar graph with data points overlaid, a box and whisker plot or a violin plot (preferably with data points included) are acceptable formats.

RESPONSE: all the figures have been replotted as a bar graph with data points overlaid.

- If $n > 30$, then the entire raw dataset must be made available either as supporting information, or hosted on a not-for-profit repository, e.g. FigShare, with access details provided in the manuscript.

RESPONSE: N/A

- 'n' clearly defined (e.g. x cells from y slices in z animals) in the Methods. Authors should be mindful of pseudoreplication.

RESPONSE: n has been defined in the methods, figure legends, as well as in Figure 1 - experimental timeline.

- All relevant 'n' values must be clearly stated in the main text, figures and tables.

RESPONSE: All relevant 'n' values have been stated in the figure legends.

- The most appropriate summary statistic (e.g. mean or median and standard deviation) must be used. Standard Error of the Mean (SEM) alone is not permitted.

RESPONSE: mean and standard deviation are used for the results, which is stated in the figure legends

- Exact p values must be stated. Authors must not use 'greater than' or 'less than'. Exact p values must be stated to three significant figures even when 'no statistical significance' is claimed.

RESPONSE: Exact p values have been added to the significant figures

Dear Professor Chen,

Re: JP-RP-2025-289464R2 "**Impact of time-restricted feeding on metabolic health and adipose tissue metabolism in aged female mice with high fat diet-induced obesity**" by Theresa Bushman, Hongming Su, and Xiaoli Chen

We are pleased to tell you that your paper has been accepted for publication in The Journal of Physiology.

Yours sincerely,

Karyn Hamilton
Senior Editor
The Journal of Physiology

IMPORTANT POINTS TO NOTE FOLLOWING ACCEPTANCE OF YOUR PAPER:

- **IMPORTANT NOTICE ABOUT OPEN ACCESS:** To assist authors whose funding agencies mandate immediate public access to published research findings, The Journal of Physiology allows authors to pay an Open Access (OA) fee to have their papers made freely available immediately on publication.

- You can help your research get the attention it deserves! Check out Wiley's free Promotion Guide for best-practice recommendations for promoting your work at: www.wileyauthors.com/eeo/guide. You can learn more about Wiley Editing Services which offers professional video, design, and writing services to create shareable video abstracts, infographics, conference posters, lay summaries, and research news stories for your research at: www.wileyauthors.com/eeo/promotion.

- If you would like to receive our 'Research Roundup', a monthly newsletter highlighting the cutting-edge research published in The Physiological Society's family of journals (The Journal of Physiology, Experimental Physiology, Physiological Reports, The Journal of Nutritional Physiology and The Journal of Precision Medicine: Health and Disease), please click this link, fill in your name and email address and select 'Research Roundup': <https://www.physoc.org/journals-and-media/membernews>

EDITOR COMMENTS

Senior Editor:

Thank you for submitting your revised manuscript. We are pleased to accept it for publication in The Journal of Physiology. Thank you for your interest in The Journal and Congratulations!